# An in vitro tumorigenesis model based on live-cell-generated oxygen and nutrient gradients

Anne C. Gilmore [1,4], Sarah J. Flaherty [1], Veena Somasundaram[2], David A. Scheiblin [3], Stephen J. Lockett[3], David A. Wink[2] & William F. Heinz [3 ✉]

The tumor microenvironment (TME) is multi-cellular, spatially heterogenous, and contains cell-generated gradients of soluble molecules. Current cell-based model systems lack this complexity or are difficult to interrogate microscopically. We present a 2D live-cell chamber that approximates the TME and demonstrate that breast cancer cells and macrophages generate hypoxic and nutrient gradients, self-organize, and have spatially varying phenotypes along the gradients, leading to new insights into tumorigenesis.

[1] Optical Microscopy and Analysis Laboratory, Office of Science and Technology Resources, Center for Cancer Research, National Cancer Institute, National Institutes of Health, Frederick, MD, USA. [2] Laboratory of Cancer Immunometabolism, Center for Cancer Research, National Cancer Institute, National Institutes of Health, Frederick, MD, USA. [3] Optical Microscopy and Analysis Laboratory, Cancer Research Technology Program, Frederick National Laboratory for Cancer Research, Frederick, MD, USA. [4] Present address: Graduate School of Biomedical Sciences, St. Jude Children's Research Hospital, Memphis, TN, USA. ✉email: will.heinz@nih.gov

Concentration gradients of soluble molecules in tissue are established by their release from and consumption by cells in combination with extracellular diffusion. These gradients influence the spatial organization and phenotypes of cells in solid tissues, including tumors[1].

Current experimental tissue models do not capture the complex spatial organization of cells and molecules, or they can be difficult to interrogate microscopically. Standard 2D cell culture does not replicate long-range (>100 μm) gradients of extracellular molecules within the tumor microenvironment (TME). Microfluidic systems can impose diffusive gradients on 2D cell cultures and are designed for microscopic interrogation[2,3], but they control only a limited number of molecules of interest, the spatial organization of cells is not the same as tissue, and the molecular gradients are not naturally cell driven. Although organoids and spheroids reflect to some extent the molecular gradients that arise in actual tissues, they vary in structure[4] and are challenging to examine and quantify at high spatial resolution with long-term live-cell microscopy[5].

Hypoxic gradients in 2D cell culture were observed in 2018 with multiple cell lines cultured separately in restricted exchange environment chambers (REECs)[6]. The gradients were cell-generated, and the cells in hypoxic regions died, analogous to necrotic zones in solid tumors. This resulted in radially symmetric cell 'disks' surrounding the chamber aperture. The cells in normoxic and low-oxygen regions had morphological variations suggesting different phenotypes depending on position along the hypoxic gradient.

Here we report the application of the REEC system to investigate the TME. Specifically, we examine the role of $O_2$ and nutrient gradients on tumor and immune cell phenotype, using 4T1 mouse mammary tumor cells and ANA-1 mouse macrophages as exemplars. The 4T1 model shares many features with triple-negative human breast cancer (TNBC)[7], and macrophages are the most abundant non-cancer cells in the TME and play an immunosuppressive role in tumorigenesis[8]. These cell types were cultured separately and in co-culture in the chamber, and the spatiotemporal dynamics of $O_2$ gradient formation, nutrient uptake, disk formation, and cell survival were quantified.

## Results

**Restricted exchange environment chamber models the tumor microenvironment.** 4T1 mouse mammary tumor cells were cultured in REECs to model the TME in an experimentally accessible system (Fig. 1a). Cells in this chamber, supported on a standard #1.5 glass coverslip (0.17 mm thick), grow in a small (<20 μL) lower compartment separated from a larger upper compartment (~1 mL) by a coverslip with a small central aperture (~0.7 mm diameter) through which $O_2$ and soluble molecules (e.g., nutrients, cytokines, and cellular waste products) diffuse between the compartments. Thus, cells directly beneath the aperture are exposed to $O_2$ and nutrients at the concentration of the upper compartment, while cells distal to the aperture exist in a cell-generated hypoxic environment. We next demonstrate the utility of the REEC system by quantifying the effects of hypoxia and nutrient gradients on 4T1 phenotype and metabolism.

**Cell-generated $O_2$ concentration gradients develop within hours.** Dissolved extracellular $O_2$ concentration ([$O_2$]) measurements in the REECs showed that radial negative gradients form rapidly (<2 h) after placement of the chamber top onto uniformly distributed (~75% confluence) 4T1 cells in 2D cell culture (Supplementary Figs. 1, 2), which agreed with mathematical modeling of the 4T1-REEC system using a diffusion-consumption model (Supplementary Fig. 3). Fluorescence labeling of the live 4T1 cells

with the Image-iT Green Hypoxia Reagent (IGHR) revealed that positive gradients of intracellular hypoxia formed in a similar timescale and showed that the hypoxic front (the distance at which the IGHR signal is 90% of its maximum) was within a millimeter of the edge of the opening after 2 h (Supplementary Fig. 1). MDA-MB-231, MDA-MB-468, and BT-549 human TNBC cell lines and mouse ANA-1 and bone marrow-derived macrophages (BMDMs) cultured in REECs also developed $O_2$ gradients within 2 h, although the amount of time required to reach equilibrium varied between cell types (Supplementary Fig. 4).

**4T1 cells cultured in REECs form stable disks.** To investigate the effect of these gradients on long-term cell culture, we used pimonidazole reduction to measure intracellular hypoxic gradients and DAPI staining to measure cell density in 4T1 cells cultured in REECs for 6 h, 2 days, 4 days, and 8 days (Fig. 1b, c). Over a period of days cell density increases in the normoxic region and decreases in the hypoxic regions of the chamber as a disk of 4T1 cells forms around the opening. Live/dead staining of 4T1 cells in REECs and live-cell imaging of GFP-expressing 4T1 cells (GFP-4T1s) in REECs revealed that disk formation is driven by a combination of cell death in the hypoxic regions, cell proliferation in the normoxic regions, and cell migration from the hypoxic regions to the normoxic regions of the chamber (Supplementary Movie 1, Supplementary Fig. 5). When the media in the upper compartment is refreshed every 2–3 days, the hypoxic gradients and disk diameters equilibrated between day 4 and day 8, and the disk diameter remains stable for at least 28 days. When the media in the upper compartment was not refreshed, the disk diameter continued to shrink until all cells were dead. In stable disks (i.e., those with little to no change in diameter over time), increased cellular hypoxia distal to the hole was measured by pimonidazole reduction and hypoxia-inducible factor 1 alpha (HIF1α) localization to the nuclei (Fig. 1d, e). Live-cell imaging of GFP-4T1s cells demonstrated that after the chamber top is removed and normoxia is re-established, the disk expands via a combination of cell proliferation and cell migration (Supplementary Fig. 6, Supplementary Movie 2).

Notably, these experiments demonstrated that 4T1 cells respond to hypoxic gradients, which do not arise in standard cell culture and cannot be readily observed microscopically in intact tumors or 3D cell culture models, by converting to a migratory phenotype in order to coalesce near the opening in the chamber. This behavior is characteristic of tumor cells in vivo and was shown here to be possible without the presence of other cell types, extracellular matrix, or 3D environment.

Hypoxia, through HIF1α stabilization, is known to induce pro-tumor phenotypes (Supplementary Fig. 7), including a collective-to-amoeboid transition in cell migration, increased in vimentin expression, and increased migration of 4T1 cells[9]. HIF1α is primarily responsible for the acute response to hypoxia, and its activity peaks within the first 24–48 h of hypoxia[10]. We observed that 4T1 cells in the periphery of the disk exhibited a temporary increase in vimentin expression relative to E-cadherin expression, between 12 and 36 h after chamber placement in low-glucose (5 mM) media and an increase after 36 h in high glucose (25 mM) media, suggesting that a transient phenotypic shift to a mesenchymal state may contribute to cellular migration during disk formation (Supplementary Fig. 8)[11]. Live-cell imaging of GFP-4T1s cells cultured in REECs demonstrated that the initiation of cell migration occurred nearly simultaneously (at 32 h) in hypoxic regions of the chamber and the cessation of migration occurred at 60 h at the periphery of the newly-formed cell disk (Supplementary Movie 1). The time difference between the measured increase in vimentin expression and the initiation of migration in the high

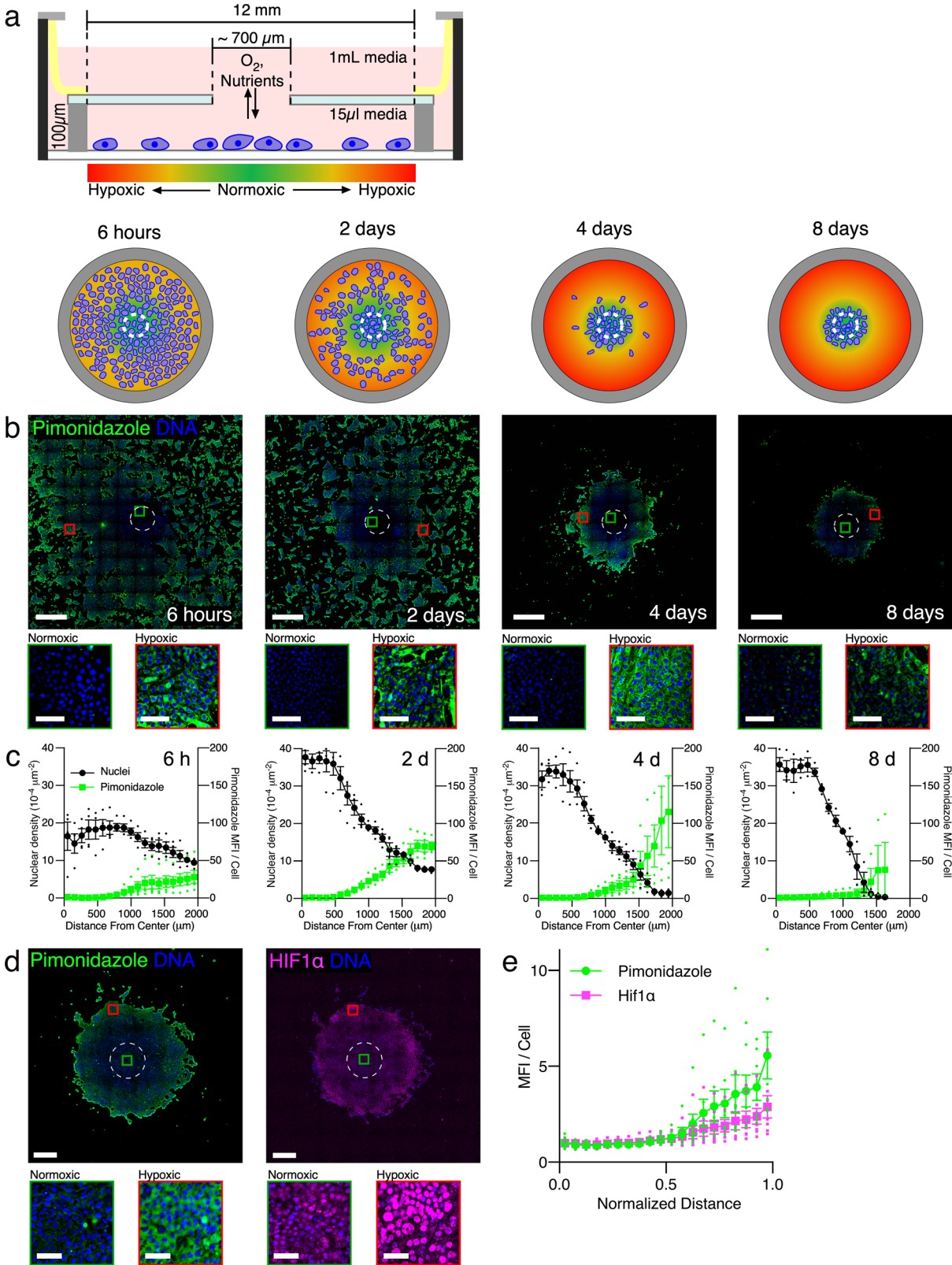

glucose experiments is likely due to small variations in chamber heights or cell plating density and thus the magnitude and temporal evolution of the cell-generated hypoxic gradients. Therefore, the REEC system shows potential as a method for investigating hypoxia-induced epithelial–mesenchymal plasticity in vitro.

**Human TNBC cells form hypoxic gradients and stable disks.** The dynamics of disk formation, which we observed in highly metastatic 4T1 cells, may vary across cell lines. Human TNBC cell lines, MDA-MB-231, MDA-MB-468, and BT-549, which are known to have differences in their response to hypoxia[12] and

**Fig. 1 The 4T1-REEC model of the tumor microenvironment (TME). a** Schematic (side view) of a restricted exchange environment chamber (REEC) and schematic (plan view) of the evolution of cell-generated hypoxic gradients and cell disk formation. **b** Widefield images of 4T1 cells in REECs generating hypoxic gradients within 6 h and forming disks centered on the $O_2$ source over 8 days. Hypoxic cells are identified by pimonidazole immunofluorescence (green), and nuclei are stained with DAPI (blue). Scale bars = 1 mm, inset scale bars = 100 μm. **c** Quantification of hypoxic gradients and cell density as a function of radial distance to the center of the chamber opening. **d** Widefield image of pimonidazole (left) and HIF1α (right) immunofluorescence staining of 7-day old 4T1 disk that developed from a uniform distribution of cells. Scale bars = 500 μm, inset scale bars = 50 μm. **e** Quantification of pimonidazole and nuclear HIF1α immunofluorescence in 4T1 disks as a function of normalized distance from opening to the edge of the disk (N = 7 disks). "Normalized Distance" refers to the relative distance from the center to the edge of a disk. MFI per cell is normalized to the first point. Dashed circles indicate the opening. MFI mean fluorescence intensity. Magnified areas of normoxic (green squares) and hypoxic (red squares) cells are displayed below each image. Data plotted as mean ± SEM.

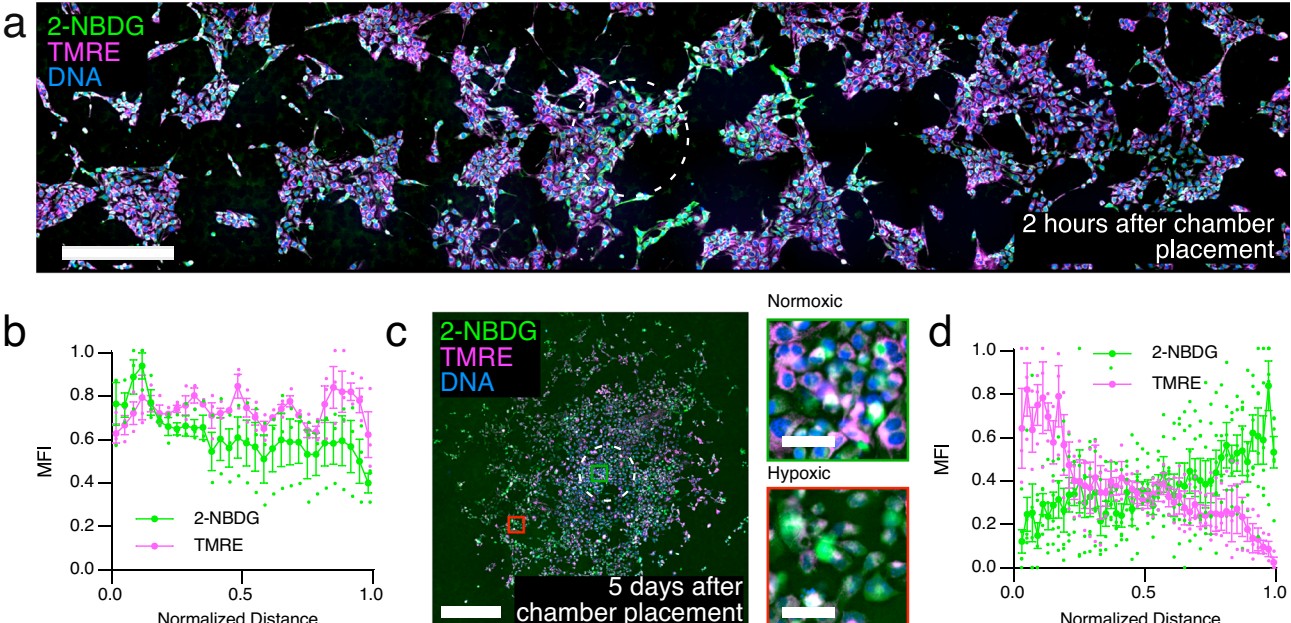

**Fig. 2 Nutrient uptake and metabolic activity vary with [O₂] concentration gradient between 2 h and 5 days in the 4T1-REEC model of the TME. a** Widefield of image of 2-NBDG uptake and TMRE fluorescence in a 4T1-REEC 2 h after chamber placement. Scale bar = 500 μm. **b** Quantification of 2-NBDG and TMRE fluorescence in 4T1-REECs 2 h after chamber placement (N = 3). Correlation coefficient of 2-NBDG, TMRE = 0.1748. **c** Widefield image of 2-NBDG and TMRE fluorescence in a 4T1-REEC 5 days after chamber placement. Magnified areas of normoxic (green square) and hypoxic (red square) cells are displayed to the right of the image. Scale bar = 500 μm, inset scale bars = 50 μm. **d** Quantification of 2-NBDG (N = 6 disks) and TMRE (N = 3 disks) fluorescence in 4T1-REECs 5 days after chamber placement. Correlation coefficient of 2-NBDG, TMRE = −0.5089. Dashed lines indicate the opening. MFI mean fluorescence intensity. Data plotted as mean ± SEM.

epithelial–mesenchymal plasticity[13], were therefore cultured in REECs. As with the 4T1 cells, the human cells rapidly formed hypoxic gradients (Supplementary Fig. 4), followed by the formation of cell disks over a period of days (Supplementary Fig. 9). However, the formation rate and size of the stable disks varied between cell lines, which we attribute to the differential oxygen consumption rates[14,15], responses to hypoxia, and epithelial–mesenchymal plasticity of the various cell lines.

**Metabolic gradients develop in 4T1 disks**. During initial [O₂] gradient formation (<2 h) glucose uptake by cells exhibited a radial gradient with maximum uptake at the opening at early time points (Fig. 2a, b). However, no gradient of mitochondrial membrane potential (ΔΨm) was observed via tetramethylrhodamine-ethyl-ester (TMRE) fluorescence 2 h after chamber placement, suggesting that oxidative phosphorylation continues for some time after hypoxia is established. In stable disks (>72 h), ΔΨm decreased while glucose consumption increased with radial distance (Fig. 2c, d; Supplementary Fig. 10), which is consistent with a metabolic shift from oxidative phosphorylation to glycolysis in the hypoxic region. Mathematical

modeling predicted no significant gradient of glucose in media containing the standard serum glucose concentration (22.5 mM), which is far in excess of the metabolic needs of the cells (Supplementary Fig. 3). Therefore, we conclude that in the case of 4T1 cells, an [O₂] gradient, but not a nutrient gradient, is required to drive the observed metabolic shift. These results demonstrate that REECs capture key features of hypoxia and metabolism in 4T1 tumorigenesis.

**Spatial distribution of NOS2 and COX2 expression and HIF1α nuclear localization in REECs is similar to that in spheroids and tissue**. We demonstrate the utility of the REEC system in an investigation of the immunosuppressive role of the inflammatory proteins inducible nitric oxide synthase (NOS2) and cyclooxygenase-2 (COX2) in the tumorigenic microenvironment. Clinically, high co-expression of these proteins in ER- human breast cancer is an indicator of very poor prognosis[16]; both are potential targets for therapy using FDA-approved anti-inflammatory drugs in combination with standard treatments. NOS2 produces nitric oxide (NO), a key regulator of cancer processes[17], from L-arginine and is upregulated by stabilized HIF1α in

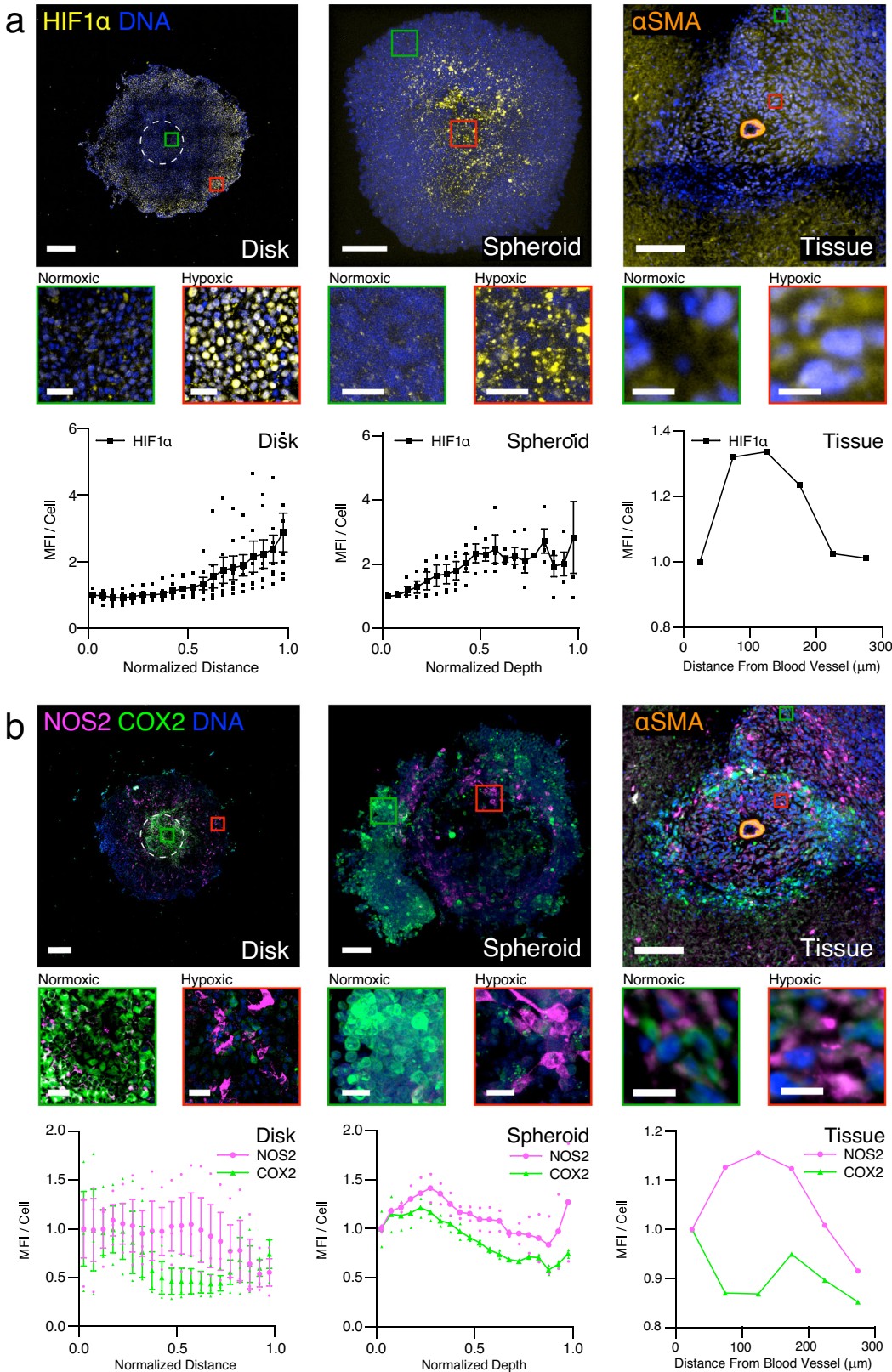

response to hypoxia and nutrient deprivation[18] (Supplementary Fig. 5). Therefore, we postulated that in the hypoxic regions of fully formed 4T1 disks, the NOS2 expression and NO flux would be high relative to COX2 expression. We observed that NOS2 levels remain high relative to COX2 in the hypoxic regions of disks, spheroids, and tissue (Fig. 3a, b; Supplementary Fig. 11).

We also saw a large increase in NO flux in the hypoxic regions of 4T1 disks, confirming our hypothesis (Fig. 4a, d, control disks; Supplementary Fig. 11).

We observed in disks, spheroids, and tumor tissue that different 4T1 cells expressed high levels of either NOS2 or COX2 and that high NOS2-expressing cells tended to be clustered

**Fig. 3 Spatial distribution of cellular phenotypes along hypoxic gradients in 4T1-REECs, 4T1 spheroids, and 4T1 tumor tissue are similar. a**
Distributions of HIF1α fluorescence in 4T1 disks ($N = 7$), spheroids ($N = 4$), and tissue ($N = 1$). **b** Distributions of NOS2 and COX2 fluorescence in 4T1
disks ($N = 3$), spheroids ($N = 2$), and tissue ($N = 1$). The tissue images capture a cross-section of a blood vessel, identified by immunofluorescent staining
of α-smooth muscle actin (αSMA) which labels the vascular smooth muscle cells. "Normalized Distance" refers to the relative distance from the center to
the edge of a disk. "Normalized Depth" refers to relative distance from the surface to the center of a spheroid. "Distance from Blood Vessel" is measured
from the center of the blood vessel. MFI mean fluorescence intensity normalized to the first point. Magnified areas of normoxic (green squares) and
hypoxic (red squares) cells are displayed below each image. Disk scale bar = 500 μm. Spheroid and tissue scale bars = 100 μm. Inset scale bars: Disk =
50 μm, spheroid = 20 μm, tissue = 10 μm. Data plotted as mean ± SEM.

together (Fig. 3b, Supplementary Fig. 12) under hypoxic
conditions. The fact that NOS2 and COX2 are highly expressed
in different cells indicates that these proteins likely drive the
expression and activity of each other via an intercellular feed-
forward mechanism[19] mediated by the release of NO and
prostaglandin E2 (PGE2)[16]. NOS2-expressing cell clumps likely
lead to significantly higher local concentrations of NO than could
arise when NOS2-expressing cells are scattered and isolated[20] and
thus augment this paracrine mechanism.

**Inhibitors modify the spatial distribution of NOS2 and COX2
expression and metabolic phenotypes of 4T1 cells in REECs.**
Inhibitors of NOS2 and COX2 block the proteins' activities,
interrupt the feed-forward loop, and reduce tumor growth
rate[16,18]. We therefore treated 4T1 disks in REECs with NOS2 or
COX2 inhibitors for 7 days to examine the effect of the inhibitors
on protein expression, NO flux, ΔΨm, and mitochondrial mass
within the disks (Fig. 4a–d).

Relative to the untreated control, treatment with the NOS2
inhibitor aminoguanidine (AG, 1 mM) resulted in a large
compensatory increase in NOS2 expression in the more hypoxic
regions of the disk, minimal NO flux, as well as increased COX2
expression in the more normoxic regions of the disk. Relative to
the untreated control, treatment with the COX inhibitor
indomethacin (Indo, 100 μM) likewise resulted in a compensatory
increase in COX2 expression in hypoxic regions of the disk,
increased NOS2 expression, and increased NO flux across the
disk. Interestingly, nuclear HIF1α levels were lower in the hypoxic
regions of Indo-treated disks relative to the controls (Fig. 4a, b).
Given that NO flux increases in the hypoxic region with
indomethacin treatment (Fig. 4c, d), this is most likely due to
the known inhibitory effect of NO on HIF1α in hypoxic
conditions[21]. These experiments demonstrate the utility of the
REEC system, because NO flux across hypoxic gradients could
not be measured in live tissue or live spheroids, which are
microscopically inaccessible.

Metabolic membrane potential (ΔΨm) was higher in normoxic
regions compared to hypoxic regions for both treatments and
controls, as expected (Fig. 4c, d). Relative to controls, ΔΨm
was lower in Indo-treated disks, indicating decreased mitochon-
drial activity in response to the higher levels of NO across the
disk, as expected. ΔΨm was higher in AG-treated disks,
particularly in the hypoxic regions, indicating a failure to switch
to anaerobic metabolism in absence of functional NOS2. In
addition, mitochondrial mass was lower in the AG-treated disks
(Fig. 4d), likely due to the lack of NO, which plays an important
role in mitochondrial biogenesis[22]. Indo-treated 4T1 disks
contained fewer cells than control disks, mirroring effects of the
anti-inflammatories in tumors (Supplementary Fig. 13)[12,13].
Together, these results show that anti-inflammatory compounds
modulate cellular phenotypes along the hypoxic gradient, which
demonstrates the utility of the 4T1-REEC system for under-
standing treatment mechanisms in conjunction with hypoxia in
the TME.

**Macrophages co-cultured with 4T1s in REECs populate the
hypoxic zone and affect 4T1 disk size.** Macrophages localize to
areas of hypoxia and necrosis in the TME where they play an
immunosuppressive role[23]. Therefore, we cultured ANA-1 mac-
rophages in REECs either alone or with 4T1 cells. The macro-
phages, unlike 4T1 cells, did not migrate towards the opening or
form disks (Fig. 5a, b), though they did generate a hypoxic gra-
dient as observed via IGHR staining (Fig. 5c, Supplementary
Fig. 4). Macrophages stimulated with the pro-inflammatory
cytokines interferon-gamma (IFNγ) and lipopolysaccharide (LPS)
express higher levels of NOS2 and thus produce more NO, which
converts them to a glycolytic pathway and decreases their mito-
chondrial metabolism and $O_2$ consumption[20]. This effect was
observed in REECs as the hypoxic front of stimulated macro-
phages was further from the hole than that of unstimulated cells
(Fig. 5c, d). When macrophages were co-cultured with 4T1s
(Fig. 5e, f) or injected through the hole onto stable 4T1 disks, the
macrophages populated the hypoxic and 4T1-necrotic regions
(Supplementary Fig. 14). These results are consistent with mac-
rophage behavior in spheroids and in vivo in which IFNγ + LPS-
stimulated macrophages infiltrate the hypoxic core of spheroids
(Supplementary Fig. 14) and hypoxic regions of tumors[24].
Interestingly, the area of stable 4T1 disks decreased following the
injection of unstimulated macrophages, whereas injection of sti-
mulated macrophages or fresh media resulted in increases in 4T1
disk area (Fig. 5g). This suggests that the stimulated macrophage-
generated increase in extracellular NO in the chamber drives
glycolysis and reduces the $O_2$ consumption of the 4T1 and ANA-
1 cells enough to push back the hypoxic front, thereby promoting
survival and proliferation of the tumor cells.

## Discussion
We developed a two-dimensional in vitro tumor model using 4T1
mouse breast cancer cells and ANA-1 macrophages based on cell-
generated hypoxic and nutrient gradients in restricted exchange
environment chambers (REECs) compatible with standard mul-
tiwell plates. In the REEC system, cell-generated radial gradients
of [$O_2$] form and stabilize around the opening in the upper
coverslip of the chamber. Disk formation has also been observed
with several fibroblast and epithelial cell lines cultured in REECs[6].
Experiments with human TNBC cell lines and macrophages show
these cells generate similar radial gradients although with dif-
ferent time and length scales, and the cancer cells form stable
disks, suggesting differential $O_2$ consumption rates (OCR),
responses to hypoxia, and EMT potential contribute to the
observed phenotypes before, during, and after disk formation.
The REEC system can thus be used to investigate these con-
tributors to tumorigenesis.

Cellular response to hypoxia is primarily regulated by HIF1α,
which in hypoxic cells stabilizes, translocates to the nucleus, and
activates a number of transcriptional programs that promote cell
survival, tumor growth, and metastasis[25]. In breast cancer cells
hypoxia-stabilized HIF1α upregulates the inflammatory protein
NOS2 in conditions of nutrient deprivation, and NO promotes

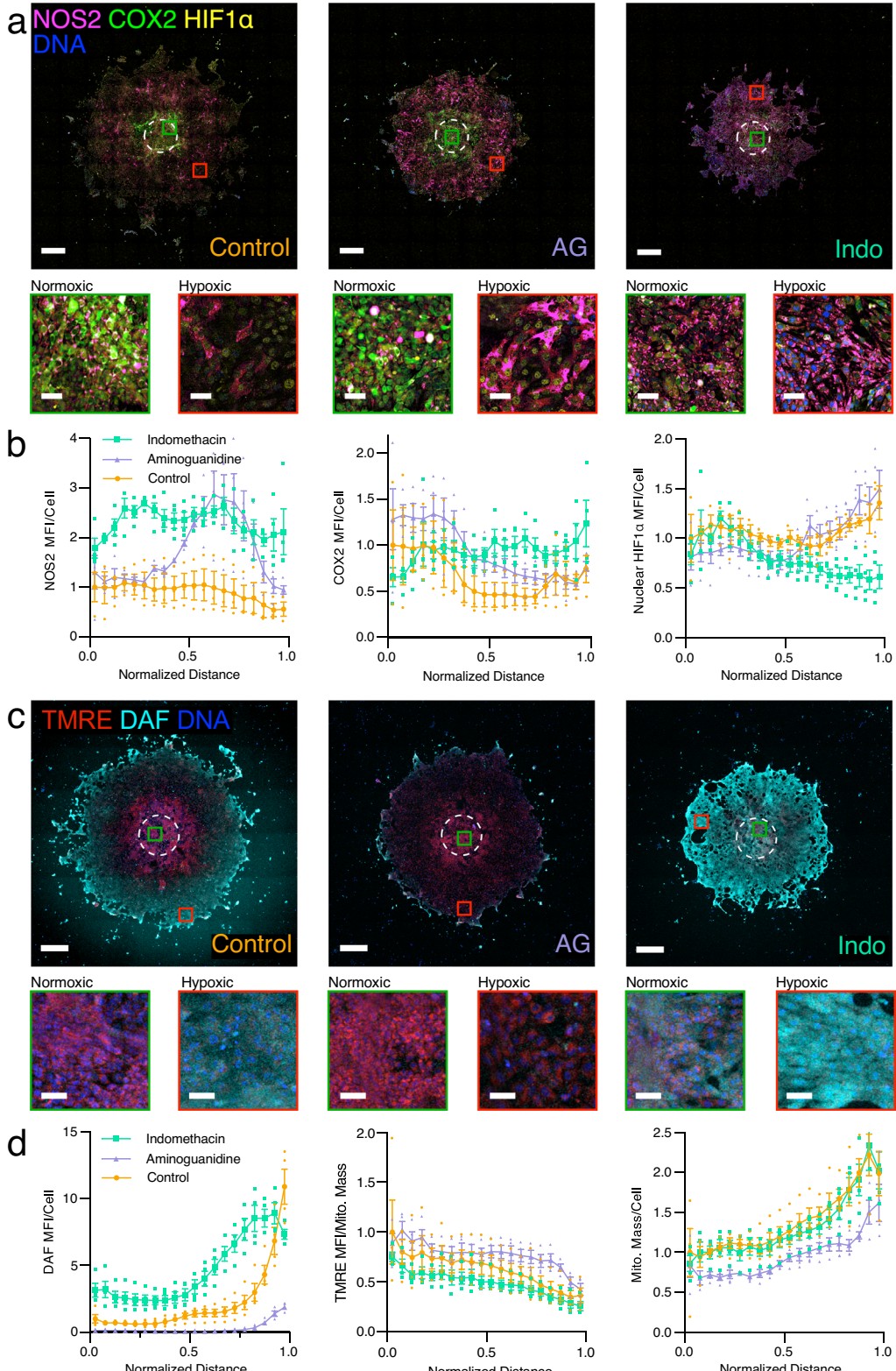

**Fig. 4 Effects of NOS2 inhibition by aminoguanidine (AG, 1 mM) or COX2 inhibition by indomethacin (Indo, 100 µM) on 4T1 phenotype along hypoxic gradients. a** Spatial distribution of NOS2, COX2, and HIF1α imaged by widefield immunofluorescence microscopy of 7-day old untreated 4T1 disks (control), AG-treated disks, and Indo-treated disks. **b** NOS2, COX2, and HIF1α distributions in 7-day old untreated 4T1 disks (control, $N = 3$), AG-treated disks ($N = 4$ for NOS2, COX2, and HIF1α), and Indo-treated disks ($N = 4$). **c** Spatial distribution of DAF and TMRE fluorescence imaged by confocal microscopy of 7-day old untreated 4T1 disks (control), AG-treated disks, and Indo-treated disks. **d** DAF, TMRE, and ATP5A distributions in 7-day old untreated 4T1 disks (control, $N = 4$), AG-treated disks ($N = 4$), and Indo-treated disks ($N = 4$). "Normalized Distance" refers to the relative distance from the center to the edge of a disk. MFI mean fluorescence intensity normalized to the first point of the untreated control. Magnified areas of normoxic (green squares) and hypoxic (red squares) cells are displayed below each image. Scale bars = 500 µm, inset scale bars = 50 µm. Data plotted as mean ± SEM.

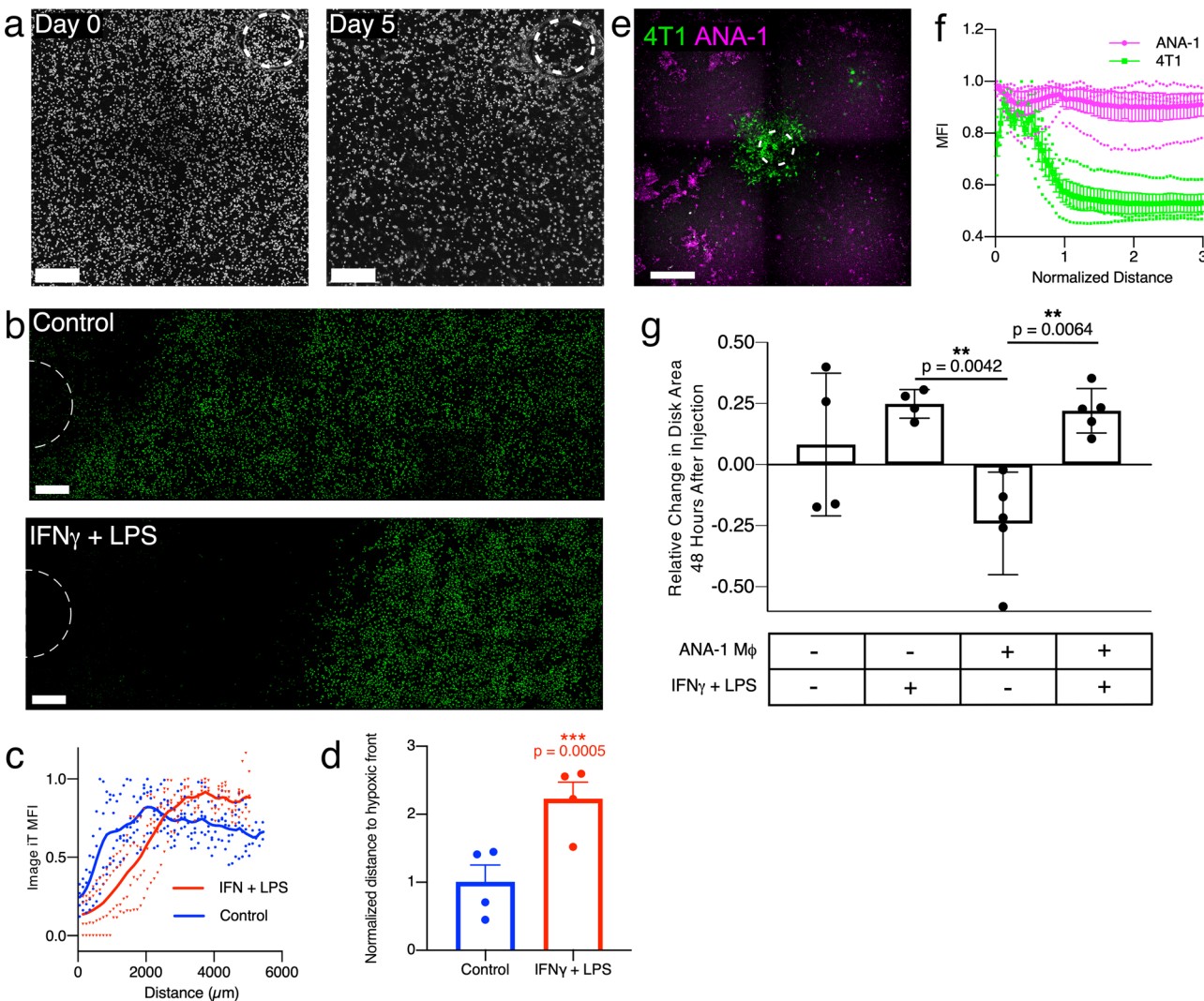

**Fig. 5 ANA-1 macrophages survive in hypoxic regions of REECs and ANA-1s stimulated with pro-inflammatory cytokines decrease oxygen consumption, both alone and in co-culture with 4T1 cells. a** Phase-contrast imaging of ANA-1s in chambers upon initial chamber placement and 5 days after chamber placement. Image contrast in inverted for display. There is very little net movement of the cells and no disks form. Scale bar = 500 μm. **b** Widefield images of Image IT Green Hypoxia Reagent (IGHR) fluorescence, in ANA-1-REECs 2 h after chamber placement, with and without 18 h of cytokine stimulation by IFNγ+LPS prior to chamber placement. Scale bar = 500 μm. **c** IGHR fluorescence intensity in ANA-1-REECs, with and without 18 h of cytokine stimulation by IFNγ+LPS, as a function of distance from the hole (data smoothed using a 5-point window to minimize oscillations resulting from a technical artifact caused by stitching of image tiles). **d** Quantification of the distance to the hypoxic front (the distance at which IGHR fluorescent reaches 90% of its maximum), normalized to the unstimulated control. ANA-1s stimulated with IFNγ+LPS increase the distance to the hypoxic front 2-fold compared to unstimulated ANA-1s (N = 4). **e** Widefield image of a 1:1 co-culture of GFP-expressing 4T1s and ANA-1 macrophages labeled with CellTracker Red CMTPX in a REEC, 7 days after chamber placement. ANA-1s persist in the hypoxic regions of the chamber where 4T1s have died or migrated toward the opening. Scale bar = 1 mm. **f** Quantification of ANA-1 and 4T1 distribution 7 days in REECs after chamber placement (N = 4). **g** Quantification of the change in GFP-4T1 disk area 48 h after injecting media with or without ANA-1s, and with or without cytokines IFNγ+LPS. Injection of unstimulated ANA-1s caused disk size to decrease because the addition of oxygen consumers to the chamber increased the size of the necrotic zone. In contrast, injection of stimulated ANA-1s caused an increase in disk size comparable to fresh media; this indicates that stimulated ANA-1s decrease oxygen consumption and thereby shrink the necrotic zone. The checkered board pattern in the images is a technical artifact due to tiling of images. Data plotted as mean ± SEM.

tumor cell survival, proliferation, and cell migration and mediates drug resistance[18]. The interactions between HIF1α and NO are complex and two-way: low levels of NO (<400 nM) can inhibit HIF1α stabilization and higher levels (>1 μM) can stabilize HIF1α in normoxic conditions[21]. We observed several downstream effects of HIF1α stabilization in the 4T1-REEC system including decreased glucose uptake, depolarization of mitochondrial membranes and adoption of a glycolytic metabolism, transient vimentin expression during disk formation, and increased NOS2 expression and NO production. The vimentin observation

suggests the 4T1 cells convert to a mesenchymal phenotype during disk formation and then revert to a more epithelial phenotype in stable disks. We will further elucidate this observation in future studies with a detailed analysis of the spatially varying cellular responses and by developing methods to integrate the REEC system with multiplexed imaging and RNA transcriptomics.

We further demonstrated that the spatial organization of 4T1 cancer cells, ANA-1 macrophages, and their phenotypes in our in vitro model are similar to those in spheroids and tissue. In the

REECs, the cell-generated [$O_2$] gradients are radially symmetric and [$O_2$] decreases with distance from the hole. This is analogous to the radially symmetric [$O_2$] gradients in spheroids, but in that 3D model, the maximum [$O_2$] is at the surface of the spheroid. In the tumor section, a hypoxic region was proximal to a nominal blood vessel, and a normoxic region was further away. At first glance this orientation is counterintuitive (although the vessel could be not functional in the tumor), and separate studies are underway to analyze the spatial relationships of vasculature and hypoxia in mouse and human tumors. However, subsequent analysis revealed that the HIF1α, NOS2, and COX2 distributions were as predicted in all three systems. In particular, we confirmed in disks the hypotheses that 4T1s express NOS2 in more hypoxic regions of the disk and demonstrated that NOS2 and COX2 are not expressed in the same cells. NOS expression is localized to a region of low-oxygen tension while COX2 expression has a complementary spatial distribution, low where NOS2 is high. Likewise, NOS2, COX2, and HIF1α expression correlated with hypoxic gradients and HIF1α expression in spheroids and tissue samples.

We acknowledge that tumor tissue samples, which comprise stroma cells, cancer cells, fibroblasts, vasculature, a host of immune cells, and extracellular matrix, are much more complex than either the REEC or spheroids. While the REEC is limited in this respect, the REEC is uniquely powerful in that it recreates intercellular molecular gradients and facilitates the analysis of the tissue environment at the individual cell level. In this study, our comparison to 3D cell culture and actual tissue has correlated well. As we continue to utilize the REEC for understanding tumorigenesis, validation of results from the REECs with results from tissue and 3D culture will be integral to these future studies. Inevitably, we will encounter results from the REECs that do not agree with actual tissue results. Such results will highlight the limitations of REECs, but more importantly they will highlight the importance of those aspects of tumors (for example the complex cellular environment and the 3D spatial nature) that are crucial in carcinogenesis.

There are several technical challenges when directly comparing REECs and spheroid models to tissue. One difficulty arises because single 5–10 μm thick tumor sections contain only fractions of individual cells, and they do not capture completely the spatial relationships within the TME. A 3D representation of the TME can be assembled from serial sections imaged with multiplexed immunofluorescence methods, and cell phenotypes and TME characteristics can be identified semi-automatically. This approach, though powerful, is time-consuming, requires significant resources, and yields a static picture of the TME. The REEC system provides a simplified, well-defined model to investigate the evolution of the spatial organization of the TME using live-cell imaging and analysis of fixed cells.

Our co-culture results suggest complex interactions between the two cell types and the hypoxic gradient. Co-cultures of 4T1 cells with macrophages mirror the spatial distribution of macrophages and tumor cells in spheroids (Fig. 5e, f; Supplementary Fig. 14) and tissue[23], in which the macrophages populate the more hypoxic and necrotic zones and the tumor cells occupy the areas closest to the oxygen source. In our experiments, the addition of macrophages to 4T1 disks in REECS initially increases the total OCR of the system and drives the decrease in 4T1 disk diameter (Fig. 5g). The difference in the disk size after co-culture with cytokine-stimulated or unstimulated macrophages can be the result of different overall OCR. Stimulation of macrophages with IFNγ leads to a metabolic switch to glycolysis, interruption of the Krebs cycle (i.e., Warburg effect), and subsequent lower OCR[26]. In standard co-cultures of GFP-4T1 cells and IFNγ + LPS-stimulated ANA-1 macrophages, the macrophages

depolarize the mitochondria of unstimulated neighboring GFP-4T1 cells via NO acting in trans[20]. Our results suggest that the decreased disk size in co-cultures with unstimulated macrophages which consume more oxygen is explained by a significant drop in oxygen tension. The recovery of disk size in co-culture with stimulated macrophages likely results from the lower OCR of the stimulated macrophages, but also the metabolic effects of macrophage-derived NO on neighboring 4T1 cells. The ANA-1/4T1 experiments are a proof of concept for co-culturing cells in the REECs to investigate short- and long-range, location dependent cell-cell interactions within the hypoxic gradients of the TME. Future studies will extend co-cultures in the REECs to include cancer-associated fibroblasts, various other immune cell types, and extracellular matrix.

In conclusion, we demonstrated that the 4T1/ANA-1-REEC in vitro model captures key features of the tumorigenic microenvironment. It recapitulates the cell-generated oxygen gradients that exist in solid tumors and via live-cell microscopy can reveal cell dynamics and phenotypes that cannot be readily determined from actual tumors. Hence, the REEC system is a powerful tool to investigate mechanisms of tumorigenesis, immunotherapy, and anti-inflammatory treatments in live cells in a tumor-like environment.

## Methods

**REEC design and assembly.** The REECs design, originally described by Hoh et al.[6], was modified to improve functionality and fit 12-well glass-bottom plates (Fig. 1a). For each REEC, a through hole (~700 μm diameter) was manually machined into a circular cover glass (18 mm diameter) using a high-speed air drill and tapered carbide bit (SCM Systems, Inc, Menomonee Falls, WI). A stainless-steel O-ring (0.100 mm thick, 12 mm ID, 18 mm OD) was epoxied to one side of the cover glass using UV-curable epoxy (Norland Products, Inc.). Laser-machined mylar clamps were epoxied to the other side of the cover glass using the same epoxy. Between each step, the epoxy was cured for 5 min in a UV-Ozone cleaner (Model 342, Jelight). Chambers were UV-sterilized immediately prior to use in cell culture (Supplementary Method).

The stainless-steel O-ring has a smoother, more uniform contact surface compared to the laser-machined Mylar gasket that was used in previously published studies. The epoxy layer between the steel spacer and the cover glass adds ~40 μm to the height of the chamber. Average chamber height was 138.4 μm ± 12.1 μm, and the interior volume was ~15.65 μL. The average hole diameter was 740.1 ± 125.1 μm.

**Cell culture and plating for the REEC system.** Murine mammary triple-negative breast cancer 4T1 cells were cultured in complete media (high glucose DMEM media supplemented with 10% fetal bovine serum (FBS), 1% penicillin, and 1% streptomycin for a final glucose concentration of 22.5 mM; Quality Biologicals). GFP-expressing 4T1 cells, 4T1-Fluc-Neo/eGFP-Puro cells (GFP-4T1, Imanis Life Sciences), were grown in complete media and selected for using G418 and puromycin. MDA-MB-231, MDA-MB-468, and BT-549 cells were obtained from Division of Cancer Treatment and Diagnosis (DCTD) Tumor Repository, Biological Testing Branch at the National Cancer Institute. The 4T1 and 4T1-Fluc-Neo/eGFP-Puro cells were validated by vendors. The MDA-MB-231, MDA-MB-468, and BT-549 cells were authenticated by DCTD and the Protein Expression Laboratory and RAS Reagents Core, Frederick National Laboratory for Cancer Research. All cell lines tested negative for mycoplasma contamination. The MDA-MB-231 cells were cultured in RPMI-1640 media supplemented with 10% FBS and 1% L-Glutamine. The MDA-MB-468 and BT-549 cells were cultured in RPMI-1640 media supplemented with 10% FBS and 1% L-Glutamine. Female BALB/c mice, aged 8–10 weeks old, were obtained from the Frederick Cancer Research and Development Center Animal Production Area. Bone marrow-derived macrophages from Balb/c mice were cultured as described previously[20] and were validated by the Wink Laboratory at NCI-Frederick. Prior to plating cells, laser-machined Mylar rims were epoxied onto a 12-well plate. The Mylar rims ensure that the Mylar clamps of each REEC are able to firmly hold the REEC chamber top against the bottom of the well. The cells were then plated (200,000 cells mL$^{-1}$, 1 mL/well) and incubated at 37 °C. Once the cells reached 70–80% confluency (~24 h), the media was refreshed and REECs were placed with sterilized forceps and pressed firmly to the bottom of the well. Media in the upper chamber was refreshed every 3–4 days with the least possible disturbance to the lower chamber.

**Mathematical model.** The REEC/4T1 system was simulated as an annulus containing a uniform density of consumers and used a diffusion-consumption model[27] to characterize the time-evolution and steady-state behavior of oxygen and glucose.

For a radially symmetric system, the concentration ($C$) of a molecular species is described by:

$$\frac{\partial C}{\partial t} = \frac{1}{r}\frac{\partial}{\partial r}\left(rD\frac{\partial C}{\partial r}\right) - n_{\text{consumers}}k_{\text{consumption}}f(C) \quad (1)$$

Where $n$ = concentration of consumers, $k$ = the per-cell consumption rate, and $f(C)$ is a function of concentration and can be written as:

$$f(C) = C_{\text{bulk}}\frac{1}{1 + \frac{C_{\text{bulk}}}{C}} \quad (2)$$

At steady state, a characteristic distance can be defined as:

$$\lambda = \sqrt{\frac{D}{n \cdot k}} \quad (3)$$

$k$, can be calculated from the maximum per-cell molecular consumption rate, $A_{\text{max}}$, and the concentration of the molecule of interest, $C_{\text{bulk}}$:

$$A_{\text{max}} = kC_{\text{bulk}} \quad (4)$$

MATLAB's partial differential equation toolbox was used to model the evolution of oxygen and glucose gradients within the REEC/4T1 system at 37 °C. The system was defined as an annulus with an inner radius, $r_1$, of 350 μm and an outer radius, $r_2$, of 6000 μm, corresponding to the radius of the REEC opening and the inner radius of the REEC, respectively. The inner radius was modeled as a source of the diffusing molecules with a constant concentration, $C_{\text{bulk}}$.

Thus, the boundary conditions were:

$$C(r_1, t) = C_{\text{bulk}} \quad (5)$$

$$\frac{\partial C(r_2, t)}{\partial r} = 0 \quad (6)$$

The initial condition corresponded to the moment the chamber top was placed in the well:

$$C(r, 0) = C_{\text{bulk}} \quad (7)$$

Oxygen diffusion, glucose diffusion, and glucose consumption rate values for 4T1 cells were based on the literature (Table 1)[28,36–39]. Oxygen consumption rate (OCR) was measured directly via a Seahorse XF96 metabolic analyzer (see below).

The oxygen concentration used in the dynamic model at $r < 350$ μm is 171 μM, not 178 μM which is the value at the interface of the media and the atmosphere. It is somewhat less because the cells immediately below the opening in the REEC consume the oxygen diffusing through the opening in the cover glass. To account for this, we modeled the oxygen concentration in a column of media 250 μm tall (100 μm high REEC and a 150 μm thick cover glass) above a monolayer of 4T1 cells using a model based on Fick's law[28]. At the top of the column, the media above the REEC cover glass is assumed to be fully oxygenated (178 μM). Using a 4T1 density was 200,000 cells cm$^{-2}$ and the $A_{\text{max}}$ for the 4T1s, the $O_2$ concentration at the cell layer was calculated to be 171 μM.

For each molecular species (oxygen or glucose), the simulation was run for an equivalent of 48 h for each combination of maximum cellular molecular consumption rate, $A_{\text{max}}$, and cell density, $n$. Convergence to steady state was defined as a change in RMS difference of <0.1% between successive profiles.

**Oxygen consumption rate measurements**. 4T1 cells' OCR was measured using the XF96 Seahorse Metabolic Analyzer (Agilent Technologies, California). 4T1s were plated ($1 \times 10^5$ cells) in each well (200 μL) of a Seahorse microplate. The plates were then incubated at 37 °C for 2 h to allow time for the 4T1 cells to adhere. Mitochondrial stress tests were performed per manufacturer's instructions. The OCR was measured as cells were treated sequentially with oligomycin (inhibitor of complex V in the electron transport chain (ETC)), trifluoromethoxy

carbonylcyanide phenylhydrazone (FCCP, Sigma-Aldrich, a depolarizer of the mitochondrial membrane potential), and rotenone and antimycin-A (inhibitors of complex I and III in the ETC, respectively). Basal respiration, ATP-linked respiration, and spare capacity were calculated using the Seahorse software.

**4T1 mouse mammary tumor model**. The NCI-Frederick Animal Facility, accredited by the Association for Accreditation of Laboratory Animal Care International, follows the Public Health Service Policy for the Care and Use of Laboratory Animals. Animal care was provided in accordance with the procedures outlined in the Guide for Care and Use of Laboratory Animals. Protocols for in vivo studies were approved by the NCI at Frederick Animal Care and Use Committee (ACUC). Female Balb/c mice obtained from the NCI at Frederick Cancer Research and Development Center Animal Production Area were housed five per cage. Eight- to ten-week-old female Balb/c mice were subcutaneously injected with $2 \times 10^5$ 4T1 cells. The allograft tumor volume was measured by Vernier caliper and calculated as volume (mm$^3$) = (width$^2$ × length)/2. When the tumors reached 2000 mm$^3$, typically 30 days post injection, the mice were euthanized, tumors were collected for analysis. Tumors were flash-frozen in liquid nitrogen and the tissues were cut into 10-μm-thick sections by the Molecular Histopathology Laboratory at NCI-Frederick.

**Fixation**. 4T1 cells cultured in 12-well plates were fixed in 4% v/v paraformaldehyde for 15 min. Samples were rinsed three times in PBS and then blocked and permeabilized in blocking buffer (3% BSA w/v, 0.3% Triton X-100 in 1× DPBS) for 1 h.

Fresh frozen sections of 4T1 tumors were fixed in 4% v/v paraformaldehyde for 30 min. Samples were rinsed three times in PBS and then blocked and permeabilized in blocking buffer for 1.5 h.

**Immunofluorescence staining**. After being fixed, blocked, and permeabilized, cultured 4T1 cells were stained with antibodies diluted in blocking buffer. Incubation times, temperatures, dilutions, and secondaries (if necessary) were used as described in Table 2. For overnight incubations, the samples were kept in a humidified chamber. Cells were then washed three times with 1× PBS and stained with DAPI (300 nM; ThermoFisher Scientific) for 15 min in 1× PBS. Cells were rinsed an additional three times with 1× PBS prior to storage or imaging.

After being fixed, blocked, and permeabilized, fresh frozen sections of 4T1 tumors were stained overnight at 4 °C with antibodies diluted in blocking buffer, as described in Table 2. Samples were rinsed three times with PBS, stained with DAPI (300 nM) for 30 min, rinsed again, and sealed for imaging on the Nikon Eclipse Ti widefield fluorescence microscope.

**Multiplexed immunofluorescence**. After imaging with appropriate filter sets for the directly conjugated antibodies (Table 2), cell and tissue samples were quenched[29,30] using a solution comprised of 1 part hydrogen peroxide (30% w/v), 1 part sodium bicarbonate (1 M, pH 10), and 3 parts purified water for 30–60 min at room temperature. Samples were then washed three times with 1× PBS and imaged to ensure quenching. Samples were re-blocked with blocking buffer as described above and re-stained with a new set of directly conjugated antibodies. The quenching cycle can be repeated for at least four rounds of staining without noted damage. DAPI does not quench and can be used to register images for processing. Imaging was performed on the Nikon Eclipse Ti widefield fluorescence microscope.

**Stimulation of ANA-1 macrophages**. A murine ANA-1 macrophage cell line was established by infection of normal bone marrow from C57BL/6 mice with J2 recombinant virus[31,32]. The cells were cultured in complete media and were stimulated via treatment with IFNγ (100 U mL$^{-1}$) and LPS (20 ng mL$^{-1}$) for 18–24 h. The media was then removed and replaced with fresh stimulation media and subsequent experiments were performed immediately.

**Injection of ANA-1 cells into 4T1-REECs**. In order to allow time for disks to form, REECs were placed on a 70–80% confluent monolayer of 4T1-GFP-luc cells, using the method described above, 3 days prior to ANA-1 injection. ANA-1 cells were stimulated with IFNγ and LPS, as described above, 1 day prior to ANA-1 injection. On the day of ANA-1 injection, ANA-1 cells were incubated in serum-free media with CellTracker Red CMTPX Dye (5 μM, ThermoFisher) at 37 °C for 30 min. The ANA-1 cells were then spun down and diluted in complete media (+/−IFNγ and LPS) to ~10,000,000 cells mL$^{-1}$. Immediately prior to injection, the media in the upper chamber of the REECs was refreshed with complete media (+/−IFNγ and LPS). The concentrated ANA-1 solution (10 μL, containing ~100,000 ANA-1 cells) was injected through the opening, directly into the lower chamber of the REEC. For controls, complete media (10 μL, +/−IFNγ, and LPS) containing no ANA-1 cells was injected. Images were taken every 2 h for 48 h on the Nikon Eclipse Ti widefield fluorescence microscope with a ×4 dry objective and using a live-cell heated stage.

**Extracellular $O_2$ concentration quantification in REECs via $O_2$ sensor foils**. To quantify the spatial variation of dissolved [$O_2$] in the media across the REEC we

**Table 1 REEC/4T1 diffusion-consumption model parameters.**

| Parameter | Value | Reference |
|---|---|---|
| $D$ (μm$^2$ s$^{-1}$) | | |
| $O_2$ | 3,370 | 36 |
| Glucose | 616 | 37 |
| $A_{\text{max}}$ (molecules cells$^{-1}$ s$^{-1}$) | | |
| $O_2$ | $2.98 \times 10^7$ | OCR measured with Seahorse XF mito-stress test with 4T1 cells. |
| Glucose | $6.47 \times 10^7$ | 38 |
| $C_{\text{bulk}}$ | | |
| $O_2$ | 178 μM, 171 μM | 28,39 |
| Glucose | 25 mM | DMEM media specification (Quality Biologicals) |

**Table 2 Antibodies and fluorophores.**

| Antibody & fluorophore | Clone | Vendor – Catalog number | Host | Dilution | Time | Temp. |
|---|---|---|---|---|---|---|
| E-cadherin, direct conjugate (AlexaFluor-488) | 24E10 | Cell Signaling Technologies – 3199S | Rabbit | 1:100 | O/N | 4 °C |
| Vimentin, direct conjugate (AlexaFluor-555) | D21H3 | Cell Signaling Technologies – 9855S | Rabbit | 1:50 | O/N | 4 °C |
| DAPI | NA | ThermoFisher | NA | 1:100 | 15 min | RT |
| NOS2, direct conjugate (AlexaFluor-568) | EPR16635 | AbCam – 209595 | Rabbit | 1:100 | 1 h or O/N | RT 4 °C |
| COX2, direct conjugate (AlexaFluor-488) | D5H5 | Cell Signaling Technologies – 13596S | Rabbit | 1:100 | 1 h or O/N | RT 4 °C |
| HIF1α, direct conjugate (AlexaFluor-647) | EPR16897 | AbCam – 208420 | Rabbit | 1:100 | O/N | 4 °C |
| ATP5A1 (Mitochondrial ATPase, Primary) | NA | AbClonal – A5884 | Rabbit | 1:100 | O/N | 4 °C |
| (Anti-rabbit CF-640R secondary for ATPase) | NA | Biotium – 20178 | Donkey | 1:100 | O/N | 4 °C |
| α-Smooth Muscle Actin, direct conjugate (eFluor-488), eBioscience | 1A4 | ThermoFisher – 41-9760-82 | Mouse | 1:400 | O/N | 4 °C |

used $O_2$ sensor foils (PreSens Precision Sensing GmbH, Germany)—100 μm thick hydrogels impregnated with particles with $[O_2]$-dependent luminescence. The ratio of the red to blue luminescence correlates with dissolved $O_2$ concentration. The $O_2$ sensor foils were glued to the inside top surface of the REEC (made with 200 μm thick stainless-steel washers to maintain a 100 μm chamber height above cells) and a hole was drilled through both the glass and the foil. The response of the sensor foils to $[O_2]$ was calibrated prior to placement on cells using glucose oxidase/catalase solutions of known $[O_2]$ using the Nikon microscope with the ×4 objective lens, using the TRITC and DAPI emission filters in rapid succession and exciting the foil with a 395 nm LED lamp. The $[O_2]$ in the glucose oxidase/catalase solution was measured using a Piccolo oxygen sensor system (PyroScience GmbH, Germany). To measure the $[O_2]$ dynamics across the REEC with 4T1 cells in phenol red-free media (1 mL), images of the sensor foil were collected every 15 min using the ×4 objective lens. To minimize effects of mixing on the live-cell $[O_2]$ measurements, the microscope stage was not moved during data collection. Thus, only a portion of the chamber was able to be imaged, and we typically collected a quadrant of the chamber with the hole in a corner of the image. A flat-field correction was applied to the images, and the red/blue intensity ratio as a function of distance from the center of the hole was converted via the calibration curve to dissolved $[O_2]$ values (Supplementary Fig. 3).

**Intracellular hypoxia gradient dynamics in REECs via Image-iT Green Hypoxia Reagent**. To measure the development of gradients of intracellular hypoxia of live cells in the REECs, cells were incubated with Image-iT Green Hypoxia Reagent (IGHR) (10 μM; Invitrogen) at 37 °C for 30 min prior to chamber placement. After chamber placement, 4T1 cells were imaged using standard FITC excitation and emission filters at ×4 or ×20 magnification on a Nikon Eclipse Ti widefield fluorescence microscope. To minimize effects of mixing on the live-cell $[O_2]$ measurements, the microscope stage was not moved during data collection, and only a quadrant of the chamber was imaged with the hole in a corner of the image. For controls, IGHR treated 4T1 cells were placed in incubators set to 0.1%, 1%, or 5% $O_2$ or in a standard incubator (~20%) for 2 h. Cells were immediately imaged (Supplementary Fig. 3).

**Intracellular hypoxia quantification in REECs via pimonidazole reduction**. To measure levels of intracellular hypoxia in cell disks, media in the upper chamber were replaced with complete media supplemented with pimonidazole (200 μM; Hypoxyprobe, Inc., Massachusetts) after disk formation. In hypoxic cells, pimonidazole is reduced and forms adducts with thiol groups. After sufficient time for the pimonidazole to diffuse through the chamber and be taken up by cells (~6 h), chambers and media were removed, and the disks were immediately fixed, blocked, and permeabilized as described above. A monoclonal antibody specific to pimonidazole adducts and conjugated with a fluorescein probe (Hypoxyprobe-Green Kit (FITC-Mab); Hypoxyprobe, Inc.) was applied for 1 h at room temperature, or overnight at 4 °C. Samples were rinsed three times with 1× PBS and imaged using standard FITC excitation and emission filters at ×20 magnification on the Nikon Eclipse Ti widefield fluorescence microscope. For controls, cultured 4T1 cells were treated with complete media which had been supplemented with pimonidazole and deoxygenated (<1% $O_2$) with glucose oxidase and catalase. HIF1α expression was determined by immunofluorescence and correlated with pimonidazole adduct formation (Fig. 1h).

**Glucose uptake gradient dynamics in REECs**. (2-(N-(7-Nitrobenz-2-oxa-1,3-diazol-4-yl)Amino)-2-Deoxyglucose (2-NBDG) (100 μM; ThermoFisher Scientific) with Hoechst (1 μg mL$^{-1}$) in low-glucose media (5 mM glucose, 10% FBS) was diffused into the chamber for 30 min at 37 °C at the desired time after chamber top placement (e.g., 2 h or 7 days). This dye was often combined with tetramethylrhodamine, ethyl ester (TMRE) (1 nM; ThermoFisher Scientific) in order to obtain simultaneous live-cell metabolic measurements. Chambers were removed, rinsed three times in 1× PBS,

and immediately imaged using standard excitation and emission filters at ×20 on the Nikon Eclipse Ti widefield fluorescence microscope or on a Zeiss 710 Laser Scanning Confocal Microscope with a ×10 dry objective.

**Live/dead staining**. Four to six days after chamber top placement, media was replaced with media containing the live/dead cell stains ethidium homodimer (2 μM), calcein AM (3 μM), and Hoechst (1 μg mL$^{-1}$; ThermoFisher) for 30 min. Cells were then imaged on the Nikon Eclipse Ti widefield fluorescence microscope using a live-cell heated stage.

**Celigo image acquisition and analysis**. Cells were cultured in 12-well plates with chambers for up to 28 days with weekly media replacement. The plate was scanned using the Cell Counting feature in the Celigo Imaging Cytometer (Nexelom Biosciences) with the brightfield algorithm to detect cells. Cells were segmented using the built-in software.

**NOS2 and COX inhibitor treatments**. Immediately prior to chamber top placement, the media in each well was replaced with treatment media: complete media supplemented with either the COX inhibitor indomethacin (100 μM) or the NOS2 inhibitor aminoguanidine (1 mM). Cells were maintained in the REECs for 3–7 days using treatment media before fixation and immunofluorescence staining. For controls, standard complete media was used.

**Nitric oxide production in REECs**. 4-Amino-5-methylamino-2′,7′-difluorofluorescein diacetate (DAF) (ThermoFisher) was used to measure and spatially resolve nitric oxide (NO) production in disks. 7 days after chamber placement, wells were washed three times in 1× PBS to remove all phenol red and BSA, which interfere with DAF fluorescence. To ensure that the lower compartment of the chamber was adequately washed, 1× PBS was gently pipetted up and down directly over the opening. The 4T1 cells were then incubated in phenol red-free, serum-free media with DAF (10 μM) and Hoechst (μg mL$^{-1}$) at 37 °C for 45 min. This dye was often combined with TMRE in order to obtain simultaneous measurements of mitochondrial membrane polarization state. Samples were then immediately imaged on a Zeiss 710 Laser Scanning Confocal Microscope with a ×10 dry objective using standard FITC emission and excitation filters.

**Metabolic gradient quantification in REECs**. Cells were incubated in phenol red-free media with TMRE at 37 °C for 20 min for cells in standard culture, or 45 min for cells in a REEC. TMRE was used simultaneously with 2-NBDG or DAF Diacetate. Cells were imaged with or without chamber removal, depending on the experiment. For controls, the electron transport chain was inhibited with FCCP or antimycin-A and rotenone. These treatments caused the TMRE signal to decrease significantly within 15 min. TMRE was imaged on a Zeiss 710 Laser Scanning Confocal Microscope with a ×10 dry objective using Texas Red excitation and emission filters.

To confirm that mitochondrial mass was consistent across the disk, disks were then fixed, blocked, and permeabilized, as described above. The disks were stained with an ATP-synthase antibody (Abclonal, A5884, Rabbit) diluted 1:100, and incubated at 4 °C overnight. Cells were rinsed with PBS and incubated with an anti-rabbit secondary at room temperature for 1 h and then imaged on a Zeiss 710 Laser Scanning Confocal Microscope with a ×10 dry objective.

**Spheroid growth, clearing, and imaging**. A spheroid formation assay was performed in ultra-low attachment round-bottom 96-well plates (Nexcelom, Lawrence, MA, USA). 4T1 cells or 4T1-GFP-luc cells (for co-culture experiments) were plated in each well (200 μL, $6 \times 10^3$ mL$^{-1}$) with serum-free DMEM supplemented with basic Fibroblast Growth Factor (20 ng mL$^{-1}$) and B-27 supplement (1:50; ThermoFisher Scientific). Media was supplemented on Day 4. Monoculture

spheroids were fixed, cleared, immunolabeled with antibodies, and imaged on Day 7 (see below). For co-culture experiments, ANA-1 macrophages were stimulated for 18 h followed by treatment with CellTracker Red CMTPX dye (10 μM) at 37 °C for 45 min and were then added to the spheroids ($2 \times 10^5$ cells/well). The ANA-1 spheroid co-cultures were fixed, cleared, immunocytochemically stained with antibodies, and imaged on Day 7 (see below).

Spheroids were cleared for imaging using the Ce3D method[33]. Briefly, spheroids were fixed in 4% v/v paraformaldehyde containing 0.5% Triton-X-100. Spheroids were blocked at 37 °C for 36 h in a humidified environment. Spheroids were then stained with directly conjugated antibodies for proteins of interest (HIF1α, NOS2, COX2) at 4 °C for 36 h. Spheroids were stained with DAPI (300 nM) for 30 min and then rinsed three times with 1× PBS. The spheroids were then embedded in 1.5% low-melt agarose. Samples were placed in clearing solution (0.1% v/v Triton X-100, 13% N-methylacetamide, 66% w/v Nycodenz AG) at room temperature for 4 h at a 1:4 agarose:clearing solution ratio. After 4 h, the clearing solution was replaced with the equivalent volume of fresh clearing solution and left overnight.

The cleared spheroids were imaged on a Leica TCS SP8 Laser Scanning Confocal microscope at ×20 with oil to match the index of refraction of the clearing solution (slightly higher than 1.5). Z-stacks were taken through the center of the spheroids.

**Image processing and data analysis**. Images taken on the Nikon Eclipse Ti widefield fluorescence microscope were typically stitched and background corrected using a rolling ball technique in Nikon Elements software, then processed in Imaris (Bitplane). Phase images, alternatively, were bandpass filtered in FIJI to remove illumination artifacts. Image brightness and contrast were adjusted to optimize the visual dynamic range for display. Imaris was used for cell segmentation and to extract position, fluorescence intensity, and geometrical statistics on each cell. The FIJI plugin StarDist[34] was used to segment nuclei in Fig. 1b. Custom R scripts were used to process the output of Imaris statistics, including average cell intensity for each channel and position. For cell disks (unless otherwise noted), cells were binned into annuli every 50 μm from the center, and the mean fluorescence intensity (MFI) per cell in that bin was calculated. Disks that could not be segmented were analyzed in FIJI using the Radial Profile plug-in, which averages the intensity value of all pixels at each radius from a fixed point. For the spheroid and tissue images, custom MATLAB functions were used to calculate fluorescence intensity and depth (spheroid) or position (tissue) for each pixel and averaged as above. For disks and spheroids intensity profiles, R scripts were used to average those radial mean values within 50-μm-wide annuli. To compare results across all stable cell disks, radial values were normalized to disk radius, and MFI was normalized to the value at $r = 0$ for that disk or the control group disk. Disk radius was measured using the circle tool in FIJI. For the NBDG/TMRE data, MFI was normalized to the maximum MFI value for each profile, and the normalized curves were averaged. For spheroids, depth values were normalized to the maximum depth of each spheroid (such that 0 represents the surface and 1 is the maximum depth), and MFI was normalized to the value at the spheroid surface. For tissue, MFI was normalized to the first position in the profile. The ANA-1 hypoxic front profiles were smoothed using a 5-point window in GraphPad Prism to minimize oscillations resulting from a technical artifact caused by stitching of image tiles. The brightness and contrast of the images were adjusted for display purposes only.

**Statistics and reproducibility**. All data are from at least three experiments and are presented as mean ± SEM unless otherwise noted. Measurements of individual cell disks quantified the fluorescence intensity and position of ~$10^4$ cells within each disk, and data from multiple (e.g., $N \geq 3$) disks per experimental condition were averaged for each analysis. Statistical significance, determined using Welch's two-tailed t-test, and Pearson correlation coefficients, was calculated in GraphPad Prism and Microsoft Excel.

**Reporting summary**. Further information on research design is available in the Nature Research Reporting Summary linked to this article.

## Data availability

The source data underlying all quantitative figures are provided as Supplementary Data 1. All other data supporting the findings of this study are available from the corresponding author upon reasonable request.

## Code availability

The code implementing the diffusion-consumption model and radial intensity profile quantification is available from GitHub (https://github.com/Will-Heinz/OMAL-Diffusion-Consumption-Simulation-and-Analysis-Tools-for-REECs) and Zenodo (https://doi.org/10.5281/zenodo.4552772)[35].

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

## Acknowledgements

This project has been funded in whole or in part with Federal funds from the National Cancer Institute, National Institutes of Health, under Contract No. 75N91019D00024 and by the Intramural Program of the NIH, NCI, Center for Cancer Research. The content of this publication does not necessarily reflect the views or policies of the Department of Health and Human Services, nor does mention of trade names, commercial products, or organizations imply endorsement by the U.S. Government. NCI-Frederick is accredited by AAALAC International and follows the Public Health Service Policy for the Care and Use of Laboratory Animals. Animal care was provided in accordance with the procedures outlined in the "Guide for Care and Use of Laboratory Animals" (National Research Council; 2011; National Academy Press, Washington, D.C.). In addition, we would like to thank Dan McVicar for the generous use of his Seahorse XF Analyzer.

## Author contributions

W.F.H., S.J.L., and D.A.W. conceived the project, and A.C.G., S.J.F., V.S., and W.F.H. designed the experiments. A.C.G. and S.J.F. constructed the REECs, performed all experiments with the chambers, wrote analysis software, and processed and analyzed the images. V.S. grew the spheroids and performed the mouse experiments and assisted A.C.G. with the Seahorse measurements. A.C.G. imaged the spheroids and tissue samples. D.A.S. assisted with multiplexed immunofluorescence imaging and image analysis. WFH implemented the mathematical model. A.C.G., S.J.F., and W.F.H. wrote the manuscript, and all authors critically reviewed the manuscript.

## Funding

## Competing interests

The authors declare no competing interests.
