## [Peer Review File · Communications Biology]

Reviewers' comments:

Reviewer #1 (Remarks to the Author):

This study describes development of an in vitro tumorigenesis model designed to produce a cell-generated gradient of nutrient and oxygen. Despite the system works with a 2D configuration it still allows studies of cancer cell behaviour in differentially developed tumour microenvironment conditions, including co-culturing systems. Hence, it provides a potential technological improvement for in vitro studies.

Overall the work appears well performed and described. There is no major concern related to the quality of the work. The major issue might relate to the capability of a 2D system to correctly recapitulate the conditions of a 3D tissue microenvironment. The attempts of the authors to make comparison with tissues still do not fully solve the issue. This should be clearly stated in the discussion. Besides that, there is still interest in the use of such model. It is therefore opinion of this reviewer that this study only requires a minor revision as follow:

Specific points:

1- Staining for Pimo and HIF-1 (Fig. 1g,h) should be made more readable and convincing. I would suggest the use of a magnification inset of the images as well as a containing with a HIF-1 target as a further control (i.e. CAIX).

2- Co-culture experiments are shown in presence of macrophages? Would would be the applicability with other co-culture systems, for example CAFs?

3- Relate to the previous point, I would give more emphasis to co-culturing application and hence I would suggest to move the experiments ANA/T4 to the main figures.

4- To facilitate the flow of the text I would suggest to move the description of the chamber (Fig. 1a) from the "introduction" to the "results" section.

5- As mentioned for figure 1, the use of magnification insets should actually be expanded to all the key data across the manuscript. This will help to more convincingly display the result.

Reviewer #2 (Remarks to the Author):

Gillmore and colleagues describe a novel 2D in vitro system to study the effect of hypoxia on tumor cell metabolism and migration. Further, they provide a new system to study the effect of macrophages on tumor cell behavior. Nevertheless, the study would further improve by the following clarifications and experiments

Major. Comments:

1. The study would greatly benefit to include other cell lines with the differential capacity to i) respond to hypoxia and nutrients and ii) undergo EMT. A minimum of three different cell lines are required for the key experiments. This is particularly important if the authors want to conclude that the O₂ availability alone drives the metabolic shift

2. The authors repeat. The. Key experiments with bone marrow-derived macrophages instead of using an artificial mouse cell line such ANA1.

3. The macrophage experiments should be moved to the main MS and figure 4 should ideally be moved to the supplement. iNOS/Cox inhibitory experiment to not add significant sufficient new knowledge or shed valuable light on the effects of oxygen on tumor cell behavior

Minor comments

The authors need to clarify when GFP+ expressing 4T1 tumor cells are used and not i.e. figure. 1e vs. 1g. It would be preferable to have the 4T1 cells to express another fluorescence marker since the hypoxia reagent is also in FITC (488 signal).

Reviewer #3 (Remarks to the Author):

In the manuscript "An in vitro tumorigenesis model based on live cell-generated oxygen and nutrient gradients" Gilmore with colleagues present the application of previously reported chamber in creation of in vitro 2D gradients. The creation of gradient systems can indeed significantly increase the physiological relevance of in vitro models and the development of reproducible and reliable techniques in this field will help to establish gradient models more widely in the scientific community.

The following concerns, however, must be considered:

1. The time points for different measurement methods are different – e.g. Figures 1c-f demonstrate the presence of gradient up to 120 min. HIF1a stabilization and pimonidazole reduction, however, is demonstrated first after 7 days in culture (Figure 1g and h). The authors should demonstrate the presence of oxygen gradients for the periods longer than 2 hours. Moreover, HIF-1a is mostly responsible for acute response to hypoxia (with its activity peaks within the first 24-48 h of hypoxia; (Wiesener, M. S. et al. Induction of endothelial PAS domain protein-1 by hypoxia: characterization and comparison with hypoxia-inducible factor-1alpha. *Blood* 92, 2260–2268 (1998)). How stable is the gradient and can a steady-state be achieved at some point in time (since we see the drop of oxygen tension over the time through the disks)?
2. The authors talk about "fully formed disks" at 72h - what criteria do they use to classify the disks as "fully formed"?
3. Figure 1e: can the authors provide the full image of the disk when using Image iT Hypoxia reagent? Can the gradient be observed over the entire disk?
4. Figure 1e vs. Figure 1g – if the scale bars are correct, the disk area becomes at least two times smaller if cells are cultivated for longer than 2 hours? Is this why the authors use "Normalized distance" in Fig 1h instead of "Radius" as shown in Fig. 1d and Fig 1f?
5. The use of GFP-expressing cells in hypoxic culture conditions is not the best choice, since molecular oxygen is believed to affect fluorophore oxidation (fluorophore maturation of GFP is oxygen-dependent).
6. The authors claim cell proliferation after reoxygenation, but relative change in disk area (Supplemental Figure 2) can also be the result of cell migration. What happens to the disk size if the chamber is not removed?
7. The authors claim that "anti-inflammatory compounds modulate cellular phenotypes along the hypoxic gradient" (Line 150). To be able to state this conclusion with certainty, experiments with indomethacin treatment in different hypoxic conditions (simple uniform culture without gradients are needed and 4T1-REEC cell viability after treatment must be studied e.g. by 21% O₂, 5% O₂, 2% O₂ and 1% O₂).

8. It is unclear if oxygen gradients retained when co-culture experiments are performed. Since both cell lines consume oxygen, it is anticipated that in such co-cultures a faster drop in oxygen occurs. Moreover, the difference in the disk size after co-culture with treated or untreated macrophages can be the result of different OCR by treated and untreated cells. It was demonstrated that treatment with IFN-gamma leads to a metabolic switch to glycolysis, interruption of the Krebs cycle ("Warburg effect") and subsequent lower OCR (Viola, Antonella, et al. "The metabolic signature of macrophage responses." *Frontiers in immunology* 10 (2019)). Thus, the decrease of the disks in co-cultures with untreated macrophages which consume more oxygen is explained by significant drop in oxygen tension.

Response to Reviewers

Dear Reviewers,

Thank you for the favorable reviews. We appreciate the positive and constructive comments. We have made substantial revisions to both the figures and text of the manuscript in order to include new experimental results and an expanded Discussion section. We have added results from several more cell lines as well as added results from new control experiments. Italicized text below is quoted from the reviewers' comments. Rather than restate our revised text in the manuscript here, we cite the revised line numbers here and highlight in the manuscript file the revised text in red font. Revised figures are presented in numerical order after the replies and within the revised manuscript. Below, we address the specific concerns of each reviewer.

Reviewer 1

We thank the reviewer for the positive comments: "*Hence, it provides a potential technological improvement for in vitro studies.*", "*Overall the work appears well performed and described. . There is no major concern related to the quality of the work*".

Major Issue

The major issue might relate to the capability of a 2D system to correctly recapitulate the conditions of a 3D tissue microenvironment. The attempts of the authors to make comparison with tissues still do not fully solve the issue. This should be clearly stated in the discussion.

We appreciate this important issue raised by the reviewer. Following the reviewer's suggestion to modify the Discussion section, we have added new a new paragraph at lines 295-303 to address this issue.

Specific points of Reviewer 1

1- Staining for Pimo and HIF-1 (Fig. 1g,h) should be made more readable and convincing. I would suggest the use of a magnification inset of the images as well as a containing with a HIF-1 target as a further control (i.e. CAIX).

We appreciate the suggestion to increase clarity through the use of magnification insets. We have therefore modified Figure 1d (formerly Fig. 1g, h) with insets to highlight the differences in Pimo and HIF1 α in the normoxic and hypoxic regions of the chamber (line 69). We added similar magnification insets to the other figures (Fig. 2, line 153, Fig. 3, line 159; Fig 4, line 197).

With respect to including HIF1 α targets in this work, we have thought of investigating downstream targets of HIF1 α and we plan to pursue this line of inquiry in detail in future efforts. With respect to this study, we have demonstrated several HIF1 α -mediated cellular responses to hypoxia that serve to validate the HIF1 α nuclear signaling. These include NOS2

An *in vitro* tumorigenesis model based on live-cell-generated oxygen and nutrient gradients. Gilmore, et al.

upregulation (Fig. 3), metabolic shifts (Figs. 2 & 4), and a transitory migratory phenotype (Supplemental Fig. 8). We added new text elaborating on these results and their relation to HIF1 α responses in more detail in the Discussion (lines 252-266).

2- Co-culture experiments are shown in presence of macrophages? What would be the applicability with other co-culture systems, for example CAFs?

This is good idea and we plan in future studies to develop additional coculture models including, for example, CAFs, T cells, and endothelial cells. In response to the reviewer's point, we have included a new section in the Discussion (lines 320-323) describing these opportunities. In the current study, though, we focused on a proof of concept that demonstrated the REEC system can be applied to multi-culture experimentation -- specifically to investigate the spatial orientation of the macrophage and tumor cell interaction and how it impacted cell behavior in the hypoxic microenvironment.

3- Relate to the previous point, I would give more emphasis to co-culturing application and hence I would suggest to move the experiments ANA/T4 to the main figures.

We appreciate the reviewer's interest in this key application of REECs. In addition to revising the text to emphasize the co-culturing applications of REECs, we have moved the ANA1-4T1-REEC results to the main text (Figure 5 and line 236).

4- To facilitate the flow of the text I would suggest to move the description of the chamber (Fig. 1a) form the "introduction" to the "results" section.

We appreciate this suggestion and moved the chamber description to the Results section (lines 62-74). In response to this comment and others, we have re-organized Figure 1 (line 69) to emphasize the evolution of the hypoxic gradients over several days. We also reorganized the Results section to describe in chronological order the development of the hypoxic gradients (lines 75-87) and the formation of the cell disks (lines 88-112).

5- As mentioned for figure 1, the use of magnification insets should actually be expanded to all the key data across the manuscript. This will help to more convincingly display the result.

We have added magnification insets throughout the manuscript (Figs 1-4), which we agree have made our results clearer and more convincing.

Reviewer #2

We thank the reviewer for the insightful and constructive suggestions.

Major Comments:

1. The study would greatly benefit to include other cell lines with the differential capacity to i) respond to hypoxia and nutrients and ii) undergo EMT. A minimum of three different cell lines

are required for the key experiments. This is particularly important if the authors want to conclude that the O₂ availability alone drives the metabolic shift

We agree that the study would greatly benefit by the inclusion of other cell lines. Our ongoing studies are progressing in this direction and in due course will lead to multiple publications about tumorigenesis expounded using the REECs. We added Supplemental Figures that include data on MDA-MB-231, MDA-MB-468, and BT549 (three TNBC human cancer lines) and murine bone marrow derived macrophages (BMDMs) (Supplemental Figs. 4 and 9). The new figures demonstrate development of hypoxic gradients in all cell types using the Image-IT green hypoxia reagent (Supplemental Figure 4 and lines 84 - 87) and disk formation of the cancer cells (Supplemental Figure 9 and lines 130-138). The BMDMs, like the ANA1 cells, did not form disks on the time scale of our studies. The gradients and disks of cancer cells form at different rates in the REECs than those of the 4T1 cells, most likely due to differences in hypoxic response as well as EMT potential. These cell-type differences and the metabolic data we present (Figs, 2 and Supplemental Fig. 10), support our conclusion that in the case of 4T1 cells, an [O₂] gradient, but not a nutrient gradient, is required to drive the observed metabolic shift.

We included the vimentin results for 4T1 cells (Supplemental Figure 8), which showed a transient increase in vimentin expression while cells are migrating to form the disk. These results demonstrate the potential application of the REEC system to the study of hypoxia induced EMT (lines 128-129). We review these results and their implications in the expanded Discussion (lines 246-251). However, we consider that an in-depth exploration of EMT within REECs incorporating multiple cell lines is beyond the scope of this paper, where we have focused on the inflammation associated proteins Nos2 and Cox2 in the 4T1 model. We appreciate the reviewer's interest in applying REECs to the study of EMT, which is the focus of a new study of ours.

We have not performed all the key experiments using three different cell lines (for example the experiments reporting the differential localization of Nos2 and Cox2) in the disk. Consequently, we have made it clear in the manuscript that key results refer specifically to 4T1 cells (e.g., lines 150-151, lines 213-216; lines 234-239, lines 259-264, lines 277-279, lines 320-322).

2. The authors repeat. The. Key experiments with bone marrow-derived macrophages instead of using an artificial mouse cell line such ANA1.

We agree with the reviewer that bone-marrow-derived macrophages may have different response in the chamber than ANA1 macrophages, and we have included an image of the hypoxic gradient formed by BMDMs in Supplemental Figure 4. We have previously cultured BMDMs in REECs to demonstrate that stimulation of the macrophages pushes back the hypoxic front (Somasundaram, et al. Redox Biology (2019), <https://doi.org/10.1016/j.redox.2019.101354>), and we include the reference in the Discussion (lines 313-315). The ANA1 [O₂] profile shown in the current work was similar to that of the BMDM in the earlier work. Here we used the ANA1 macrophages since their flux of NO and behavior is well established.

3. The macrophage experiments should be moved to the main MS and figure 4 should ideally be moved to the supplement. iNOS/Cox inhibitory experiment to not add significant sufficient new knowledge or shed valuable light on the effects of oxygen on tumor cell behavior

We appreciate this suggestion, and we have moved the macrophage results into the main body of the manuscript (new Fig. 5, line 235). We decided to keep the inhibitory experiment (Fig. 4, line 197) in the main text because with that data in combination with the new control experiment requested by Reviewer 3 (comment 7), we can now conclude more confidently that “anti-inflammatory compounds modulate cellular phenotypes along the hypoxic gradient” (lines 213-216). The data also establish the important proof-of-concept that the REEC system can be used to explore treatment mechanisms (e.g., NO flux modulation) in relation to hypoxic gradients in the TME (lines 194-196).

Minor comments

The authors need to clarify when GFP+ expressing 4T1 tumor cells are use and not i.e. figure. 1e vs. 1g. It would be preferable to have the 4T1 cells to express another fluoresce marker since the hypoxia reagent is also in FITC (488 signal).

We thank the reviewer for identifying a potential source of confusion. The use of GFP expressing 4T1 cells was limited to the live-cell disk formation, chamber top removal and regrowth, and ANA1-4T1-REEC experiments. In order to increase clarity, these results were removed from Figure 1. We modified the text and captions such that the abbreviation ‘GFP-4T1’ is used only when describing long-term, live-cell experiments with the GFP-expressing 4T1 cells (the disk formation and disk expansion videos and the ANA1-4T1-REEC experiments; lines 94-95 and 104, line 122, lines 313-315; line 352). All other references to 4T1 cells are to cells not expressing GFP.

Reviewer #3

We thank the reviewer for the very positive comment: “*The creation of gradient systems can indeed significantly increase the physiological relevance of in vitro models and the development of reproducible and reliable techniques in this field will help to establish gradient models more widely in the scientific community.*”

The following concerns, however, must be considered:

1. The time points for different measurement methods are different – e.g. Figures 1c-f demonstrate the presence of gradient up to 120 min. HIF1a stabilization and pimonidazole reduction, however, is demonstrated first after 7 days in culture (Figure 1g and h). The authors should demonstrate the presence of oxygen gradients for the periods longer than 2 hours. Moreover, HIF-1a is mostly responsible for acute response to hypoxia (with its activity peaks within the first 24-48 h of hypoxia; (Wiesener, M. S. et al. Induction of endothelial PAS domain protein-1 by hypoxia: characterization and comparison with hypoxia-inducible factor-

1alpha. Blood 92, 2260–2268 (1998)). How stable is the gradient and can a steady-state be achieved at some point in time (since we see the drop of oxygen tension over the time through the disks)?

We appreciate the suggestion to include additional time points to bridge the gap between short- and long-term measurements of hypoxic gradients in the chambers. We modified Figure 1 (line 69) to include data showing the presence and evolution of the hypoxic gradients from 6 h to 7 d after the chamber is placed on the cells. These additional results are the backbone of the revised Fig. 1 and described in the Results section (lines 62-93). Combined with the 2 h data (moved to Supplemental Figure 1), the new results demonstrate that the hypoxic gradient forms within hours after chamber application, then the gradient evolves slowly as the cell disk forms, after which it equilibrates. The new data also include quantification of cell density in the chamber as a function of time. We added the Wiesener, et al, reference to the text (lines 116-117), and we include in the Discussion more detailed analysis of the gradient evolution during the 7 days (lines 246-266).

2. The authors talk about “fully formed disks” at 72h - what criteria do they use classify the disks as “fully formed”?

We thank the reviewer for requesting this clarification. We replaced the terminology “fully formed disks” with “stable disks” throughout the manuscript, and we define a stable disk as one with “little to no change in diameter” over time (lines 101-102). We have considered 4T1-disks to be stable once the confluent region of cells surrounding the chamber aperture reaches ~120% of the stable diameter observed in long-term disks. This is typically the case by 72 hours after chamber placement on the 4T1s.

3. Figure 1e: can the authors provide the full image of the disk when using Image iT Hypoxia reagent? Can the gradient be observed over the entire disk?

For the live-cell measurements of gradient formation we imaged a single field of view in order to accurately capture the formation of hypoxic gradients within the chamber. Imaging the entire chamber, even with a low-magnification 4X objective, requires translating the stage to collect multiple fields of view for a tiled image. The stage motion would disrupt the cell-generated diffusive gradients by mixing. This is now stated in the methods section (lines 508-511 and 520-522). In order to demonstrate the formation of stable, radially-symmetric oxygen gradients across the disk, we used immunofluorescence staining of pimonidazole adducts in fixed cell disks (revised Figure 1).

4. Figure 1e vs. Figure 1g – if the scale bars are correct, the disk area becomes at least two times smaller if cells are cultivated for longer than 2 hours? Is this why the authors use “Normalized distance” in Fig 1h instead of “Radius” as shown in Fig. 1d and Fig 1f?

We appreciate the comment and agree that it is necessary to clarify the figure. The scale bars are correct, but our description may not have been sufficiently clear. The chamber is placed on

a confluent monolayer of cells, which form a disk over approximately 72 hours. Two hours after chamber placement, the cells remain a confluent monolayer, but the hypoxic gradient has been established.

To clarify the main points of Figure 1, we have modified Figure 1 (line 69) to show the evolution of the hypoxic gradients and the distribution of cells in the chamber over time (6 hours to 7 days). The new data show that the cells in the near-confluent monolayer migrate or die as the disk forms. The 2 hour time points depicting the rapid formation of hypoxic gradients over uniform monolayers of 4T1 cells (original Figs 1c-1f) have been moved to Supplemental Figure 1 (and lines 80--84).

We use normalized radius to simplify comparisons of the spatial distributions of phenotypes between disks and between disks and spheroids. A second benefit derives from the observation that the hypoxic gradients (measured by pimonidazole) monotonically increase in the disks and spheroids (Figure 1 and Figure 3). Thus the normalized radius can be used as a proxy for hypoxia in analyses where we do not measure hypoxia directly. We have also revised our methods section in order to increase clarity regarding the use of “Normalized Distance” versus “Radius” (lines 658-661).

5. The use of GFP-expressing cells in hypoxic culture conditions is not the best choice, since molecular oxygen is believed to affect fluorophore oxidation (fluorophore maturation of GFP is oxygen-dependent).

We thank the reviewer for this concern. We used the GFP-4T1 cells to observe the dynamic disk formation (lines 94-97 and 121-125) and disk expansion (lines 103-106) processes and to highlight the long-range cell motions involved. We observe the same disk formation dynamics using live-cell phase imaging with both 4T1 and GFP-4T1 cell lines. But in our experiments the intensity of the GFP is more-or-less uniform across the disk, and we do not see an increase in GFP fluorescent intensity when the chamber is removed. The new Figure 1 data which quantifies radial cell density during disk formation shows that there is a decrease of cell number in the hypoxic regions as the disk forms (line 69). Combined with the GFP live-cell data, this argues that the disk formation dynamics we see in GFP-expressing cells is due to cell migration and cell death, not a decrease in GFP fluorescence.

6. The authors claim cell proliferation after reoxygenation, but relative change in disk area (Supplemental Figure 2) can also be the result of cell migration. What happens to the disk size if the chamber is not removed?

We appreciate these insights. The change in disk size after reoxygenation is due to both cell migration and cell proliferation. We now include a live-cell time-lapse movie (Supplementary Video 2) using GFP-4T1 cells that which demonstrates that disk growth after chamber removal is due to a combination of proliferation and cell migration. If the chamber is not removed, then disks will remain stable for at least 28 days when the media in the upper compartment is periodically refreshed. We have clarified this in the Results section (lines 100-101 and 103-105).

7. The authors claim that “anti-inflammatory compounds modulate cellular phenotypes along the hypoxic gradient” (Line 150). To be able to state this conclusion with certainty, experiments with indomethacin treatment in different hypoxic conditions (simple uniform culture without gradients are needed and 4T1-REEC cell viability after treatment must be studied e.g. by 21% O₂, 5% O₂, 2%O₂ and 1%O₂).

We thank the reviewer for the suggestion, and we performed the requested control experiment. We found that cell viability decreased with O₂% for every indomethacin concentration applied over 24 h. However, for any given O₂%, there was no decrease in viability after 24 h, and only about a 25% decrease after 7d, for all indomethacin concentrations. We have included this data in Supplemental Figure 14 and the Results section (lines 207-213). We added this logic to the discussion preceding our conclusion in line 150 (now lines 213-216).

8. It is unclear if oxygen gradients retained when co-culture experiments are performed. Since both cell lines consume oxygen, it is anticipated that in such co-cultures a faster drop in oxygen occurs. Moreover, the difference in the disk size after co-culture with treated or untreated macrophages can be the result of different OCR by treated and untreated cells. It was demonstrated that treatment with IFN-gamma leads to a metabolic switch to glycolysis, interruption of the Krebs cycle (“Warburg effect”) and subsequent lower OCR (Viola, Antonella, et al. "The metabolic signature of macrophage responses." *Frontiers in immunology* 10 (2019)). Thus, the decrease of the disks in co-cultures with untreated macrophages which consume more oxygen is explained by significant drop in oxygen tension.

We completely agree with the explanation by the reviewer of our results from the co-culture experiments, and therefore we have stated these points in the Discussion section. NO from the activated macrophages affects the metabolisms of both the macrophages and 4T1 cells. We previously showed that activated macrophages bolster the intercellular movement of NO (Somasundaram et al. *Redox Biol.* 2020). Those results informed our co-culture experiments and interpretation of the current data. We have expanded the Discussion section to include a more detailed explanation and interpretation based on the reviewer’s comments (lines 308-323) and included the Viola et al reference (line 313).

Figure 1. The 4T1 REEC model of the tumor microenvironment (TME).

- a. Top. Schematic (side view) of restrictive exchange environment chamber (REEC). Bottom, Schematic (plan view) of evolution of cell-generated hypoxic gradients and cell disk formation.
- b. 4T1 cells in REECs generate hypoxic gradients within 6 h and form disks centered on the O₂ source over 8 d. Hypoxic cells are identified by pimonidazole immunofluorescence (green), and nuclei are stained with DAPI (blue). Scale bars = 1 mm, inset scale bars = 100 μ m.
- c. Quantification of hypoxic gradients and cell density as a function of radial distance to the center of the chamber opening.
- d. Widefield image of pimonidazole (left) and HIF1 α (right) immunofluorescence staining of 7-day old 4T1 disk that developed from a uniform distribution of cells. Scale bars = 500 μ m, inset scale bars = 50 μ m.
- e. Quantification of pimonidazole and nuclear HIF1 α immunofluorescence in 4T1 disks as a function of normalized distance from opening to the edge of the disk (N=7 disks). “Normalized Distance” refers to the relative distance from the center to the edge of a disk. MFI per cell is normalized to the first point.

Dashed circles indicate the opening. MFI = mean fluorescence intensity. Magnified areas of normoxic (green squares) and hypoxic (red squares) cells are displayed below each image. Data plotted as mean \pm SEM.

Figure 3. Spatial distribution of cellular phenotypes along hypoxic gradients in 4T1-REECs, 4T1 spheroids, and 4T1 tumor tissue are similar.

- a. Distributions of HIF1 α fluorescence in 4T1 disks (N=7), spheroids (N=4), and tissue (N=1).
- b. Distributions of Nos2 and Cox2 fluorescence in 4T1 disks (N=3), spheroids (N=2), and tissue (N=1).

The tissue images capture a cross section of a blood vessel, identified by immunofluorescent staining of α -smooth muscle actin (α SMA) which labels the vascular smooth muscle cells. "Normalized Distance" refers to the relative distance from the center to the edge of a disk. "Normalized Depth" refers to relative distance from the surface to the center of a spheroid. "Distance from Blood Vessel" is measured from the center of the blood vessel. MFI = mean fluorescence intensity normalized to the first point. Magnified areas of normoxic (green squares) and hypoxic (red squares) cells are displayed below each image. Data plotted as mean \pm SEM. Disk scale bar = 500 μ m. Spheroid and tissue scale bars = 100 μ m. Inset scale bars: Disk = 50 μ m, spheroid = 20 μ m, tissue = 10 μ m.

Figure 4. Effects of Nos2 inhibition by aminoguanidine (AG, 1 mM) or Cox2 inhibition by indomethacin (Indo, 100 μ M) on 4T1 phenotype along hypoxic gradients.

- a. Spatial distribution of Nos2, Cox2, and HIF1 α imaged by widefield immunofluorescence microscopy of 7-day old untreated 4T1 disks (control), AG-treated disks, and Indo-treated disks.
- b. Nos2, Cox2, and HIF1 α distributions in 7-day old untreated 4T1 disks (control, N = 3), AG-treated disks (N = 4 for Nos2, Cox2, & HIF1 α), and Indo-treated disks (N = 4).
- c. Spatial distribution of DAF and TMRE fluorescence measured by confocal microscopy of 7-day old untreated 4T1 disks (control), AG-treated disks, and Indo-treated disks.
- d. DAF, TMRE, and ATP5A distributions in 7-day old untreated 4T1 disks (control, N = 4), AG-treated disks (N = 4), and Indo-treated disks (N = 4).

“Normalized Distance” refers to the relative distance from the center to the edge of a disk. MFI = mean fluorescence intensity normalized to the first point of the untreated control. Magnified areas of normoxic (green squares) and hypoxic (red squares) cells are displayed below each image. Data plotted as mean \pm SEM. Scale bars = 500 μ m, inset scale bars = 50 μ m.

Figure 5. ANA1 macrophages survive in hypoxic regions of REECs and do not form disks, both alone and in co-culture with 4T1 cells. ANA1s stimulated with pro-inflammatory cytokines push back the hypoxic front, decrease oxygen consumption, and infiltrate the hypoxic core of spheroids.

- Phase contrast imaging of ANA-1s in chambers upon initial chamber placement and 5 days after chamber placement. Image contrast is inverted for display. There is very little net movement of the cells. Scale bar = 500 μ m.
- Widefield images of Image IT Green Hypoxia Reagent (IGHR) fluorescence, in ANA1-REECs 2 hours after chamber placement, with and without 18 hours of cytokine stimulation by IFN γ +LPS prior to chamber placement. Scale bar = 500 μ m.
- IGHR fluorescence intensity in ANA1-REECs as a function of distance from the hole (data smoothed using a 5-point window in GraphPad Prism to minimize oscillations resulting from a technical artefact caused by stitching of image tiles).
- Quantification of the distance to the hypoxic front (the distance at which IGHR fluorescent reaches 90% of its maximum), normalized to the unstimulated control. ANA1s treated with IFN γ +LPS increase the distance to the hypoxic front 2-fold compared to untreated ANA1s (N = 4).
- Widefield image of a 1:1 co-culture of GFP expressing 4T1s and ANA1 macrophages labeled with Cell Tracker Red CMTPX, 7 days after chamber placement. ANA1s persist in the hypoxic regions of the chamber where 4T1s have died or migrated toward the opening. Scale bar = 1 mm.

- f. Quantification of ANA1 and 4T1 distribution 7 days after chamber placement (N = 3).
- g. Quantification of the change in GFP-4T1 disk area 48 hours after injecting media with or without ANA1s, and with or without cytokines IFN γ +LPS. Injection of unstimulated ANA1s caused disk size to decrease, because the addition of oxygen consumers to the chamber increased the size of the necrotic zone. In contrast, injection of stimulated ANA1s caused an increase in disk size comparable to fresh media; this indicates that stimulated ANA1s decrease oxygen consumption and thereby shrink the necrotic zone.

The checkered board pattern in the images is a technical artefact due to tiling of images. Data plotted as mean \pm SEM.

Supplemental Figure 4. Human TNBC cells and mouse macrophages generate hypoxic gradients in REECs. Widefield images of mouse 4T1 cells, human TNBC cells (MDA-MB-231, MDA-MB-468, and BT-549), mouse macrophages (ANA1 and bone marrow-derived macrophages) stained with Image IT Green Hypoxia Reagent (IGHR) reveal hypoxic gradients form in the REECs within 2 h. The cells were uniformly present in the chambers, but only hypoxic cells fluoresce. The variation in distance to the hypoxic front likely results from differential oxygen consumption rates of the cell lines. Dashed circle indicates the opening. Scale bars = 500 μm .

The checkered board pattern is a technical artefact due to tiling of images. Scale bar = 500 μm .

- b.** Expression of vimentin in low glucose (5 mM) media as a function of distance from opening at different time points (N = 1 per timepoint). Vimentin expression is transiently increased in the hypoxic regions of the disk from 24 to 31 hours during disk formation.
- c.** Expression of vimentin in high glucose (25 mM) media as a function of distance from opening at different time points (N = 1 per timepoint). The 12 h timepoint is low because vimentin is an endpoint EMT marker, and at 12 h EMT has not yet begun. At later times the vimentin expression is elevated.
- d.** Ratio of maximum vimentin to e-cadherin expression in disks for different glucose concentrations over time (N = 3 disks per time point per concentration). The e-cadherin levels did not vary appreciably across the REEC over time. In the 5 mM case, after peaking during disk formation, the ratio returns to the initial value in stable disks. Data plotted as mean \pm SEM.

Mean fluorescence intensity (MFI) is normalized to cell number at each distance from the opening.

REVIEWERS' COMMENTS:

Reviewer #1 (Remarks to the Author):

The revision is fully satisfying. I am happy to endorse the article for publication.

Reviewer #2 (Remarks to the Author):

The authors have improved the paper greatly. I have no further comments.

Reviewer #4 (Remarks to the Author):

The authors conducted all the necessary additional experiments and meticulously answered the questions I asked. Additional data obtained by the authors supplemented the manuscript and provided new information.

However, I must point the wrong choice of cell viability evaluation under hypoxia (Sup. Fig. 14). MTT assay uses mitochondrial activity to convert MTT into formazan, but mitochondrial activity is decreased under hypoxia. That means that MTT assay is not suitable for determining cell viability/cell survival rate after hypoxia. That would explain why we do not see any difference in cell viability after the treatment with indomethacin. What the authors are measuring here is mitochondrial activity. The authors should better remove these data as well as text discussing it from the manuscript.

Response to Reviewers

Dear Reviewers,

Thank you for your very positive comments on our revised manuscript. We address the remaining concern of Reviewer #4 below.

Reviewer 4

We thank the reviewer for a detailed reading of the revision.

Major Issue

However, I must point the wrong choice of cell viability evaluation under hypoxia (Sup. Fig. 14). MTT assay uses mitochondrial activity to convert MTT into formazan, but mitochondrial activity is decreased under hypoxia. That means that MTT assay is not suitable for determining cell viability/cell survival rate after hypoxia ... The authors should better remove these data as well as text discussing it from the manuscript.

We appreciate the argument about the MTT assay. We have removed Supplemental Figure 14 and the associated discussion and methods sections as suggested.